# The myosin X motor is optimized for movement on actin bundles

Virginie Ropars[1,2,*], Zhaohui Yang[3,*], Tatiana Isabet[1,2,*], Florian Blanc[1,2,4], Kaifeng Zhou[5], Tianming Lin[3], Xiaoyan Liu[3], Pascale Hissier[1,2], Frédéric Samazan[1,2], Béatrice Amigues[1,2], Eric D. Yang[3], Hyokeun Park[6], Olena Pylypenko[1,2], Marco Cecchini[4], Charles V. Sindelar[5], H. Lee Sweeney[3,**] & Anne Houdusse[1,2,**]

Myosin X has features not found in other myosins. Its structure must underlie its unique ability to generate filopodia, which are essential for neuritogenesis, wound healing, cancer metastasis and some pathogenic infections. By determining high-resolution structures of key components of this motor, and characterizing the *in vitro* behaviour of the native dimer, we identify the features that explain the myosin X dimer behaviour. Single-molecule studies demonstrate that a native myosin X dimer moves on actin bundles with higher velocities and takes larger steps than on single actin filaments. The largest steps on actin bundles are larger than previously reported for artificially dimerized myosin X constructs or any other myosin. Our model and kinetic data explain why these large steps and high velocities can only occur on bundled filaments. Thus, myosin X functions as an antiparallel dimer in cells with a unique geometry optimized for movement on actin bundles.

[1] Structural Motility, Institut Curie, PSL Research University, CNRS, UMR 144, F-75005 Paris, France. [2] Sorbonne Universités, UPMC Univ Paris 06, Sorbonne Universités, IFD, 4 Place Jussieu, 75252 Paris, France. [3] Department of Pharmacology and Therapeutics and the Myology Institute, University of Florida College of Medicine, PO Box 100267, Gainesville, Florida 32610-0267, USA. [4] Laboratoire d'Ingénierie des Fonctions Moléculaires (ISIS), UMR 7006 CNRS, Université de Strasbourg, F-67083 Strasbourg Cedex, France. [5] Department of Molecular Biophysics and Biochemistry, Yale University, New Haven, Connecticut 06520, USA. [6] Department of Physics, Division of Life Science, and State Key Laboratory of Molecular Neuroscience. The Hong Kong University of Science and Technology, Clear Water Bay, Kowloon, Hong Kong. * These authors contributed equally to this work. ** These authors jointly supervised this work. Correspondence and requests for materials should be addressed to C.S. (email: charles.sindelar@yale.edu) or to H.L.S. (email: Lsweeney@ufl.edu) or to A.H. (email: anne.houdusse@curie.fr).

Class X myosin has been found to be localized at the tips of filopodia[1,2], which are plasma membrane protrusions containing bundled actin that are necessary for cellular processes such as cell adhesion, migration, angiogenesis and the formation of cell–cell contacts. Myosin X is required for filopodia formation and extension[1–3]. In fact, the motor activity of myosin X dimers even without the cargo-binding domain is sufficient for the initiation of filopodia[4]. Its unique ability to form these actin outgrowths allows myosin X to perform such functions as driving neuron extensions[2,5,6], tumor invasion[7–11], wound healing[9] and a subset of pathogen infections[9]. Thus, understanding the unique adaptations of this myosin class that enable its unique functions will provide fundamental insights into important cellular processes.

Myosin motors move along actin filaments via a series of conformational changes in the motor domain that are coupled with the sequential release of MgATP hydrolysis products, $P_i$ and MgADP. These conformational changes in the motor domain are amplified by the myosin lever arm, which is comprized of the C-terminal subdomain of the motor domain (known as the converter) and an extended alpha helix length that varies with the myosin class[12]. In the case of myosin X, the lever arm contains three alpha-helical calmodulin (CaM)-binding sites and a stable, single alpha helical (SAH) domain[13,14], as characterized with a monomeric (813–909) fragment (Fig. 1a) and by measurements of monomers and dimers of myosin X heavy meromyosin (HMM) constructs from rotary shadowed electron microscopy (EM) images. Study of a short fragment (883–934) has shown that part of this SAH region can be involved in a short anti-parallel dimerization coiled-coil[15], but it is unclear if this dimerization occurs in the context of the normal flanking sequences.

Class X myosins, like those of class VI and VII, appear to exist in cells primarily as monomers[16–18] and have been proposed to dimerize only upon interaction of their tail with cargo[16,19]. It is unclear if *in vivo* dimerization occurs upon interaction with all cargos, or a subset of cargos. For myosin X, dimerization would appear to be necessary for its role in filopodia[4]. Not only are myosin motors targeted within the cell via their tail domains, but the specialized cellular function of different myosin classes require specializations of the kinetics of their motors[20], as well as differing lever arm structures[21,22]. This importance of lever arm design is just beginning to be fully appreciated[21–23], and has been the subject of controversy for class VI myosins[22–24]. In the case of myosin VI, the converter subdomain of the motor has an unusual rearrangement that allows for a different amplitude and orientation of the stroke than is seen for myosin V or myosin II[25], and the lever arm of dimeric myosin VI is extended by unfolding of a three-helix bundle, which occurs only upon dimerization[26]. Altogether, these structural features result in a myosin VI dimer step size that is similar to that of myosin V, although of opposite directionality. The myosin X lever arm, like that of myosin VI, contains fewer IQ motifs than myosin V, but its lever arm is extended by a SAH domain[13,14] (Supplementary Fig. 1a).

A tail-less myosin X dimer influences the actin filament organization to promote initiation of filopodia[4] by an unknown mechanism. Furthermore, whether or not myosin X dimers are optimized for trafficking on actin bundles in order to serve its role in filopodia remains an open question. This especially is the case since all of the myosin X constructs studied to date have been truncated and artificially dimerized in differing positions (Supplementary Table 1). Exogenous coiled-coils have been added to truncated constructs to stabilize dimers for *in vitro* single-molecule studies. This however likely perturbs the normal geometry of the full-length dimer and influences its ability to move on actin bundles and filaments, as illustrated by the very

different motility properties found for these chimeras[27–31] (Supplementary Table 1).

To begin to address these issues and understand how myosin X performs its unique cellular roles, we determined structures of the dimerization domain of myosin X, as well as the motor domain with and without the IQ motifs of its lever arm. These structures allowed us to create a model for the myosin X dimer. We then performed single-molecule studies on actin filaments and bundles with full-length myosin X (with no exogenous coiled-coil), and with a truncated, zippered dimer, that was designed based on our structure of the dimerization region. What these show is that myosin X is able to walk on actin bundles with higher velocity and larger step sizes than it does on single actin filaments. Our zippered dimer, unlike all previously studied dimers, recapitulates the single-molecule behaviour of the full-length dimeric myosin X, as well as displays the velocity of myosin X *in vivo*[32]. These data, together with specific features of the motor, as revealed by several X-ray structures and a cryo-EM reconstruction of the myosin X rigor state, provide strong evidence that myosin X has a larger lever arm movement (powerstroke), a long lever arm with flexible regions, and an antiparallel coiled-coil that optimize its movement on actin bundles in cells.

## Results

**Structure of the dimerization region of myosin X.** We first investigated whether or not the coiled-coil region is capable of forming an anti-parallel dimer, as previously described[15], in the context of longer native fragments of myosin X. If myosin X moves optimally on actin bundles, then the motility is likely highly dependent on the way the two heads are articulated. It is thus critical to describe the hinges and geometry of these elements that control how the two heads are spaced and 'gating' one another. (Gating refers to the ability of an attached rear head of a dimer to stall the lead head in a state that prevents it from completing its transitions on actin until the lead head detaches[33]). We have obtained crystals of this critical region that contain the IQ3, SAH and dimerization region (L786-A932) (Supplementary Fig. 1b). From the resulting structure at 3.5 Å resolution (Fig. 1b, Supplementary Table 2), it is clear that the SAH (E814-A846) and the following helical region (E847-E884) together form a 10.5 nm long single α-helix and elongates the IQ region. The structure also reveals how the two SAH regions are joined together by an anti-parallel coiled-coil (Fig. 1c). The boundary between the SAH and the region that participates in the anti-parallel coiled-coil is not evident from the sequence (Fig. 1d), but is evident in the new structure (Fig. 1d), and results in an SAH that is shorter than suggested for a monomeric construct[14]. The anti-parallel coiled-coil that is formed is in agreement with the NMR structure[15] (Fig. 1c).

The structure and dynamics of the SAH-CC-SAH dimeric region is particularly important to allow inferences as to the lever arm rigidity, which will dictate the space that a lead head can explore while the rear head is attached to actin. Molecular dynamic simulations of this region (Fig. 1e; Methods) show that no melting of the SAH occurs on the simulation timescale, although bending (or twisting) is possible within the SAH (Supplementary Fig. 2a–c). Interestingly, the simulation promotes a spontaneous relaxation of the SAH-CC-SAH structure that stretches this region by 1.7 nm relative to the X-ray structure, reaching a value of 25.7 nm (Fig. 1f).

**Single-molecule motility of human myosin X.** Our full-length myosin X construct fused the monomeric green fluorescent protein (GFP) derivative, mWasabi, to the N terminus of full-length human myosin X (Supplementary Fig. 1b, termed as

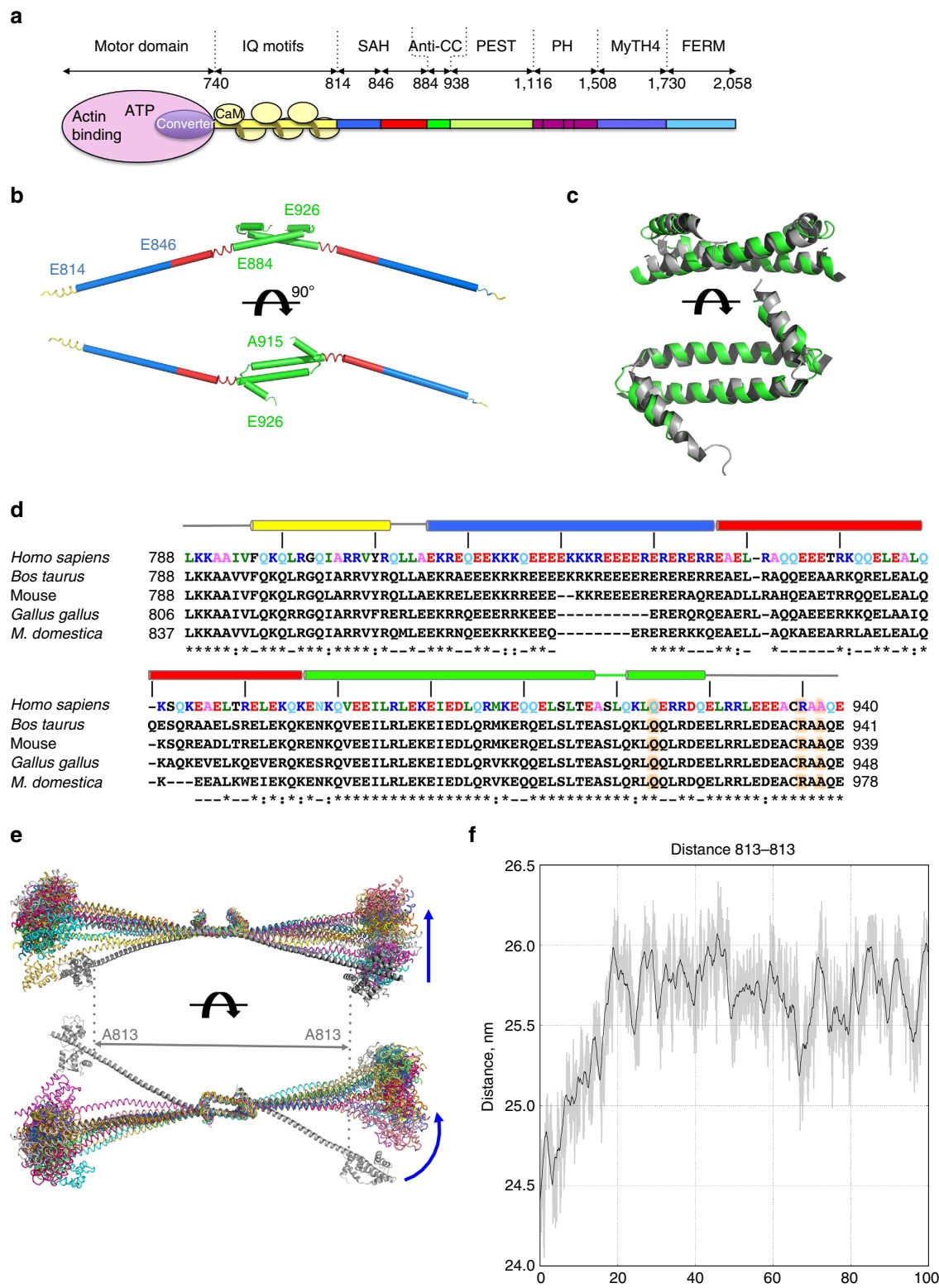

**Figure 1 | Myosin X dimerization region. (a)** Blueprint of the myosin X motor. **(b)** X-ray model of IQ3-SAH-CC. Yellow = IQ3, blue = SAH region, green = anti-parallel coiled-coil. **(c)** The dimerization region of the IQ3-SAH-CC structure (green) is compared with the short antiparallel coiled-coil structure (grey, 2LW9)[15]. **(d)** Sequence alignment of different myosin X for the IQ3-SAH-CC region. Note the variability for the SAH region. The region (E847-E884) includes a number of hydrophobic residues, unlike the more proximal portion of the SAH[13]. Note that the boundary between the SAH and the dimerization region cannot be predicted from the sequence. The IQ3-SAH-CC structure shows that the region (E847-E884) forms a SAH, rather than possibly being part of the dimerization region, as studies of myosin X chimeras would suggest (Supplementary Table 1). **(e)** Structures explored during the MD simulations of the IQ3-SAH-CC/CaM complex. **(f)** Variation of the distance between residues 813 of chains A and B during the MD simulation, describing the end-to-end distance of the SAH-coiled-coil region. The black line is the moving average with a 125 ps window; the grey envelope shows the actual values. The time-series shows that after ∼20 ns, the distance stabilizes at 25.7 nm on average, with a s.d. of 0.22 nm over the last 80 ns.

mWasabi-MyoX-FL). In addition, we engineered a truncated dimeric construct that allowed the use of a Q-dot in place of the cargo-binding tail for single-molecule studies, providing better time and spatial resolution than is possible with GFP derivatives. We created a zippered dimer (so-called HMM, Supplementary Fig. 1b) based on the structure of the myosin X dimerization region (Fig. 1b). A 19 amino-acid linker was introduced after alanine 938 to allow sufficient space between the ends of the anti-parallel coiled-coil and the leucine zipper to allow both to form without physical constraints (Supplementary Fig. 3).

As shown in Table 1, both the human myosin X full length (Supplementary Fig. 4) and zippered dimer constructs move processively on single F-actin filaments and on fascin-bundled F-actin filaments. In all cases, the distribution of step sizes is broad, with frequent backward steps, but the distribution on bundles is even broader and contains extremely large steps (Fig. 2). The broad step-size distribution was best fit to two single Gaussians on a single actin filament, with the peak of the forward steps at $36 \pm 14$ and $39 \pm 13$ nm for the full-length and zippered dimers, respectively, but the difference was not significant (Fig. 2a,c). The behaviour on bundles was similar for both constructs, but the distribution was broader and shifted towards larger step sizes. The broad distribution of forward steps on bundles were fit by Gaussians centered on $19 \pm 7$, $38 \pm 7$ and $52 \pm 5$ nm for the full-length dimer, and on steps of $17 \pm 6$, $40 \pm 9$ and $57 \pm 2$ nm for the zippered dimer (Fig. 2b,d). The differences in step sizes were not significant (see Supplementary Note). It also appeared that the reverse steps may be composed of multiple peaks, but the data was under-sampled and did not allow accurate fitting of these peaks. However, the fit for the backward steps of the HMM on bundles was possible and slightly better if fit by two populations ($49 \pm 5$ and $20 \pm 11$ nm) as shown in Supplementary Fig. 5d,e. The population of large steps measured for either the full length or the zippered dimer on actin bundles were larger than have been previously reported[27–31]. The step size in single actin filaments and the velocity of movement ($\sim 36$ nm steps and $\sim 300$ nm s$^{-1}$) were essentially the same as in some previous reports for artificially zippered dimers[29,31] ($\sim 36$ nm steps and $\sim 200$–300 nm s$^{-1}$ velocity). However, unlike the previous data[29], the full length and zippered dimers of myosin X moved faster and with longer run lengths on fascin-bundled F-actin filaments, than on single filaments (Table 1 and Supplementary Fig. 6). In fact, the velocity of movement on the bundles (660 nm s$^{-1}$) was twice as fast as on a single filament (312 nm s$^{-1}$) and the run length was more than twice as long. Note that the velocity on bundles closely matches what is observed in filopodia[32], but is more than twice as fast as any previously published values for zippered dimers[27–31] (Table 1; Supplementary Fig. 6; Supplementary Table 1).

**Kinetics of myosin X.** We measured a steady state ATPase of $\sim 9.0$ s$^{-1}$ per head for our zippered dimer (HMM) and $\sim 19$ s$^{-1}$ per head for a single-headed construct (S1: MD-3IQ-SAH) that was truncated just before the dimerization region (Supplementary Fig. 7a). (The activity of our single-headed construct was slightly

higher than the rate previously reported for a MD1IQ/CaM construct[34]). Our data reveal that our two-headed (HMM) construct is gated in the actin-activated ATPase assay, since the rate per head of the dimer on a single actin filament is roughly half that per head as measured for the single-headed construct.

Next, we addressed the possibility that some kinetic step(s) of the myosin X differ on single actin filaments as compared with bundles, in order to explain the higher velocity of movement we observe on bundles. We examined the product (phosphate and ADP) release steps, one of which is normally the rate-limiting step for a given myosin in its ATPase cycle. In agreement with the earlier kinetic characterizations[34], we measured a maximal rate of $108 \pm 4$ per second for phosphate release when S1 was first mixed with MgATP and then mixed with $50 \mu$M actin (Supplementary Fig. 7b). Since this rate is too fast to limit the ATPase activity or movement, we next measured ADP release using both the S1 and HMM constructs, on both single actin filaments and on bundles. All experiments gave a mantADP release rate of $\sim 60$ per second (Table 2 and Supplementary Fig. 7c). This rate is also too fast to be limiting, and did not differ between single filaments and bundles.

Lastly, we examined the transition that occurs between the release of phosphate and the release of ADP, which is measured by quenching of pyrene-actin fluorescence, and historically has been thought to monitor the transition from weak to strong binding of myosin on actin[35]. We recently showed that the rate of this step is typically around 30 per second[36] for a variety of myosin classes and speculated that it does not monitor the formation of the first strong-binding state, but instead monitors the formation of a myosin state with strong affinity for actin and ADP[37]. This rate is slower for myosin X than for other myosins: ($\sim 19$ per second) for the S1 construct on either filaments or bundles and for the HMM construct on single actin filaments (Table 2 and Supplementary Fig. 7d). However, this rate is greatly accelerated on actin bundles for the HMM construct ($34 \pm 4$ per second). As this represents a $\sim 2$-fold acceleration of the rate limiting step in the ATPase cycle, this can account for the greater velocity of the HMM construct on actin bundles, as opposed to filaments. (Note that the data in Supplementary Fig. 7d also provides further evidence of gating).

**Structure of the myosin X motor domain.** Given the surprising large step size that we measured for the myosin X dimers, we next evaluated whether the motor domain was capable of generating a larger angular swing than myosin V. We had previously shown that myosin VI has an unusual rearrangement of its converter domain in its pre-powerstroke state (PPS)[25]. This state in fact corresponds to the structure populated by myosin upon its re-attachment to the filament with a primed (pre-stroke) position for its lever arm. We thus crystallized the motor domain of myosin X in the pre-powerstroke state (with ADP vanadate bound) at 1.8 Å resolution, as well as that of the MD2IQ of myosin X at 3.1 Å resolution (Fig. 3a, Supplementary Table 2).

The PPS structure differs in the orientation and position of the converter as compared with other plus-end myosins (Fig. 3b,e).

**Table 1 | Run length and velocity of myosin X on single F-actin and fascin-bundled F-actin filaments.**

| Construct/actin track | Run length (μm) | | | | Length constant (λ) | | Velocity (μm s$^{-1}$) | | | |
|---|---|---|---|---|---|---|---|---|---|---|
| | Mean | s.d. | s.e. | N | λ | s.e. | Mean | s.d. | s.e. | N |
| mWasabi-MyoX-FL/single F-actin | 0.81 | 0.51 | 0.03 | 283 | 0.60 | 0.19 | 0.31 | 0.08 | 0.02 | 160 |
| MyoX-HMM-A938/single F-actin | 0.64 | 0.71 | 0.03 | 768 | 0.32 | 0.08 | 0.41 | 0.20 | 0.01 | 768 |
| mWasabi-MyoX-FL/bundled F-actin | 1.95 | 1.71 | 0.1 | 285 | 1.16 | 0.24 | 0.66 | 0.20 | 0.01 | 285 |
| MyoX-HMM-A938/bundled F-actin | 1.80 | 2.11 | 0.07 | 889 | 1.18 | 0.13 | 0.68 | 0.55 | 0.05 | 103 |

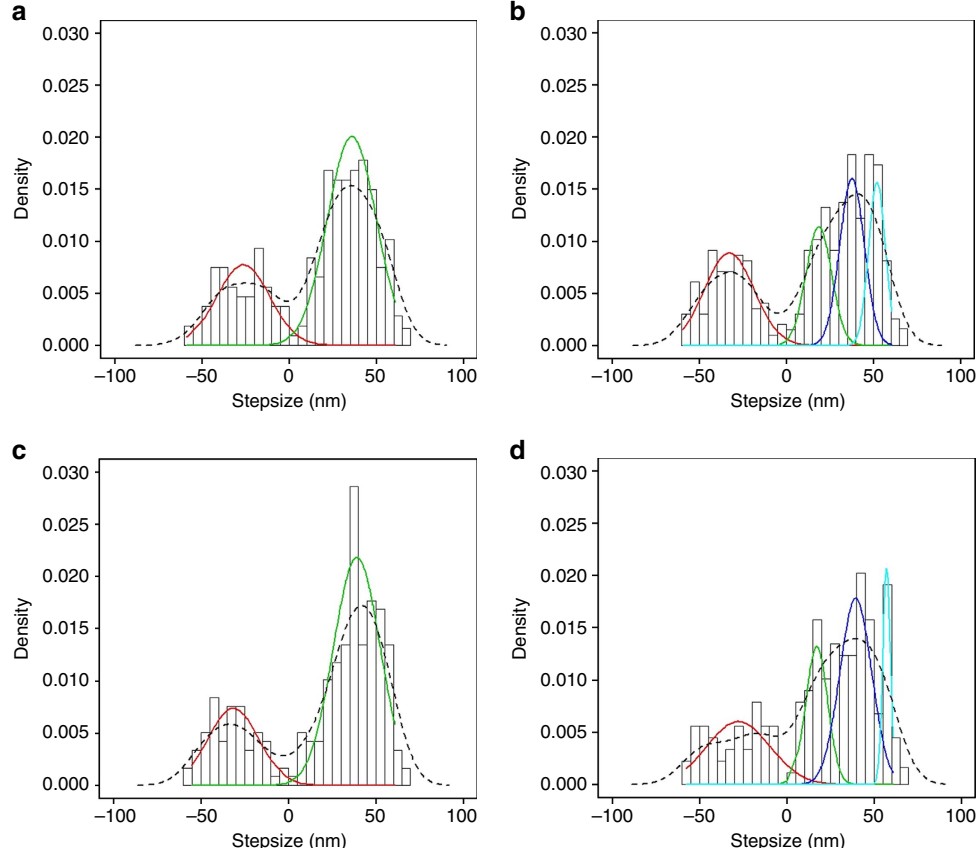

**Figure 2 | Histogram and density plots of myosin X full length and HMM step-size data.** Histograms (open vertical bars) and kernel density estimate plots (dashed curves) generated by the density function of the step-size data of myosin X full length and HMM are presented along with fitting components of multiple Gaussian distributions (solid curves; See details of the fitting procedure in Supplementary Fig. 5). (**a**) Myosin X full length on single F-actin filaments. Two Gaussian component distributions were fitted. The values for the best two Gaussian fits are $-26 \pm 15$ (s.d.) nm, and $36 \pm 14$ (s.d.) nm, $N = 214$. (**b**) Myosin X full length on fascin-bundled F-actin filaments. Four Gaussian component distributions were best fitted. The fitted values for four Gaussian components are: $-33 \pm 14$ (s.d.) nm, $19 \pm 7$ (s.d.) nm, $38 \pm 7$ (s.d.) nm and $52 \pm 5$ (s.d.) nm, $N = 393$. (**c**) Myosin X HMM on single F-actin filaments. The values for the best two Gaussian fit are $-33 \pm 15$ (s.d.) nm, and $39 \pm 13$ (s.d.) nm, $N = 238$. (**d**) Myosin X HMM on fascin-bundled F-actin filaments. Best fits for four Gaussian component distributions were obtained. The values for the Gaussian components are $-28 \pm 18$ (s.d.) nm, $17 \pm 6$ (s.d.) nm, $40 \pm 9$ (s.d.) nm and $57 \pm 2$ (s.d.) nm, $N = 178$. For the HMM on a dimer, the backward steps were best fit as two populations (Supplementary Fig. 5e), with steps of $-49 \pm 6$ (s.d.) nm and $-20 \pm 11$ (s.d.) nm.

**Table 2 | Kinetic parameters of myosin X on single F-actin and fascin-bundled F-actin filaments.**

| Construct/actin track | mantADP release (s⁻¹) | | | Δ Pyrene fluorescence (s⁻¹) | | |
|---|---|---|---|---|---|---|
| | Mean | s.d. | N | Mean | s.d. | N |
| S1 (motor domain + 3 IQs + SAH)/single F-actin | 61 | 3 | 9 | 18 | 3 | 9 |
| MyoX-HMM-A938/single F-actin | 57 | 4 | 9 | 19 | 4 | 9 |
| S1 (motor domain + 3 IQs + SAH)/bundled F-actin | 59 | 2 | 9 | 17 | 3 | 9 |
| MyoX-HMM-A938/bundled F-actin | 60 | 4 | 9 | 34 | 4 | 9 |

Interestingly, myosin X class specific interactions between the converter and the N-terminal subdomain (Nter) of the motor domain (Fig. 3d, Supplementary Fig. 8a,b), that are not seen for myosin Ic (ref. 38), myosin II (ref. 39) and myosin VI (ref. 25) PPS structures, stabilize the position of the last helix of the converter and largely controls the rest of the lever arm orientation. These conserved interactions maintain the converter in the same orientation in the motor domain (MD)

and MD2IQ crystal structures despite different packing environment and the presence of the IQ region.

To compare the structure of this pre-powerstroke state with that of class V myosins, we attempted to crystallized myosin Va in this state but were unsuccessful. However, a data set was obtained from crystals of the motor domain of myosin Vc with two molecules in the asymmetric unit (Supplementary Table 2). These two myosin Vc molecules have quite similar structures for their motor domain but differ in the orientation of the converter (Fig. 3e). In contrast to myosin X, lack of interaction allows the myosin Vc converter to explore slightly different orientations (Fig. 3e).

Comparison of these PPS structures show that while the subdomains of the motor are totally conserved in position among the myosin PPS structures obtained to date (Supplementary Fig. 8c), the myosin X converter is rotated compared with that of myosin Vc (Fig. 3e) and other myosins. This is not due to a conformational change within the converter as found in myosin VI[27] (Supplementary Fig. 8d). Instead, a sharper turn at the end of the SH1 helix, (at Gly672), allows formation of specific interactions between the converter and the Nter subdomain in particular with class-specific loops of this subdomain (Supplementary Fig. 8a,b).

**Rigor state of myosin X on actin**. To complete the assessment of the angular displacement of myosin X from pre-powerstroke to rigor, we performed cryo-EM on actin filaments decorated with the motor domain of myosin X (Supplementary Fig. 1b).

The resolution of the reconstruction was 9 Å (see Methods and Supplementary Fig. 9a), which was sufficient to resolve most alpha-helical elements in the map. The conformation of myosin X thus observed (Fig. 3f) is indistinguishable from the X-ray

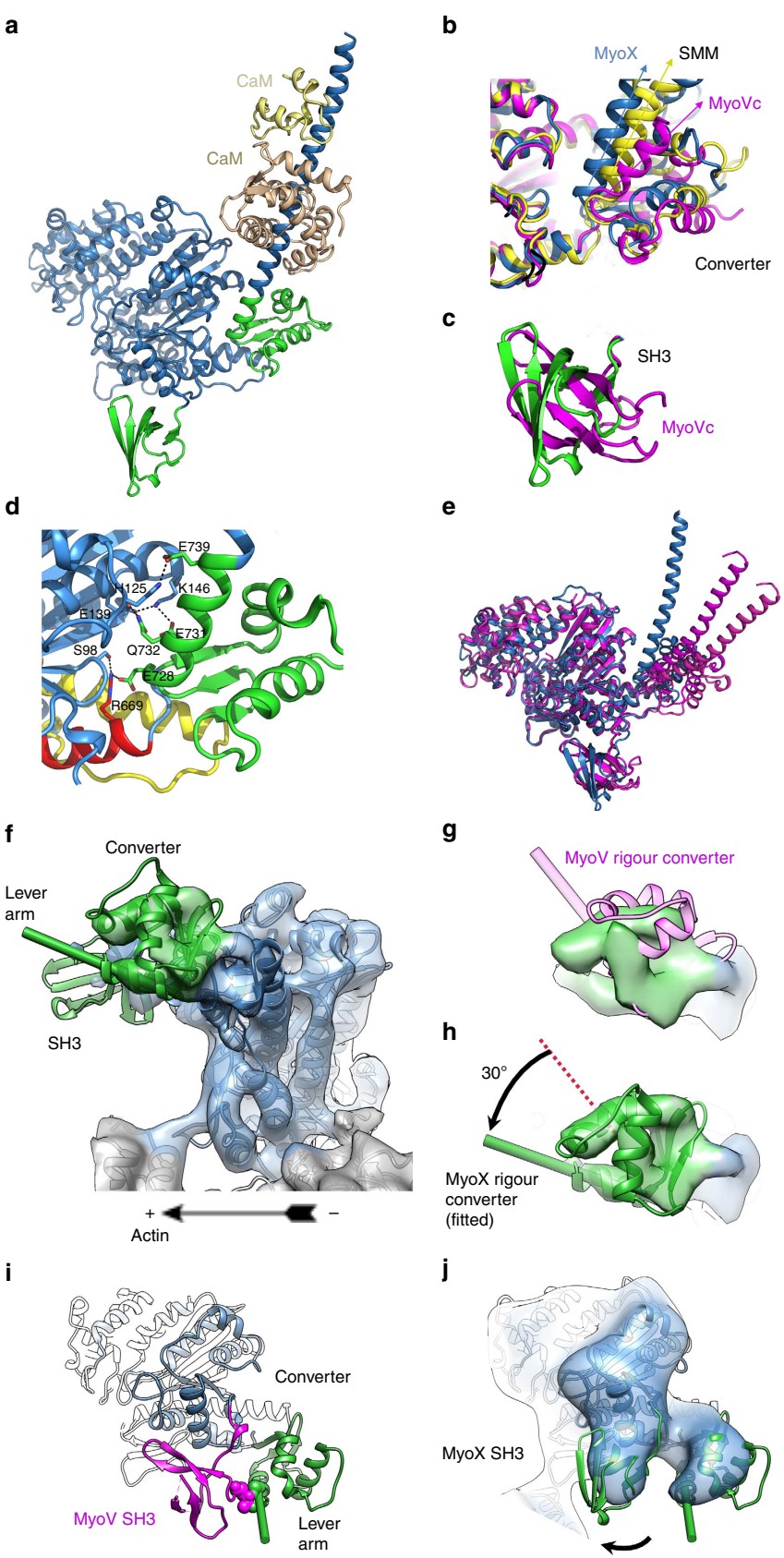

structure of nucleotide-free myosin V (1OE9 (ref. 40)), which closely represents the rigor state of myosin V on the actin filament[41]. The orientation of the myosin X motor domain on actin in our cryo-EM maps resembles that seen in previous cryo-EM reconstructions of several myosins in the rigor state[41,42].

Comparison with these acto-myosin rigor structures show, however, a substantial re-orientation of the converter in the rigor myosin X cryo-EM map (Fig. 3g,h, Supplementary Fig. 9b–d, and Supplementary Movies 1 and 2). The three short α-helices characteristic of the converter fold were directly resolved, indicating that the converter assumes a novel orientation in the rigor state of myosin X to within a cone with a half-angle of ∼5°, as assessed by rigid-body fits performed on a pair of independently refined maps derived from separate halves of the data (Supplementary Fig. 9b–d and Supplementary Movie 2). This analysis indicates that the lever arm extends toward the

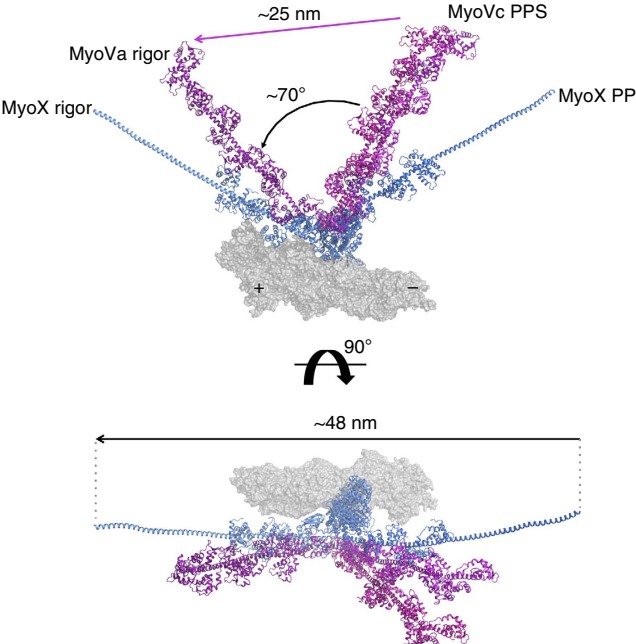

**Figure 4 | Power strokes of myosin X and myosin V derived from X-ray crystallography and cryo-electron microscopy.** Comparison of the powerstrokes of myosin V and myosin X motors in two orientations, using U50 and L50 subdomains for superimposition. Note that the powerstroke do not have a significant azimuthal component. The stroke at the end of 6 IQ of myosin V is 25 nm, while that for myosin X at the end of 3IQ-SAH is ∼48 nm.

pointed end of actin nearly parallel to the actin axis, thus differing from the orientation of the rigor myosin V lever by ∼30° (Figs 3h and 4).

Additionally, we observed a density feature corresponding to the distinctive N-terminal SH3 subdomain of myosin X (Fig. 3a,c,j). This latter feature did not exhibit obvious secondary structure in our density map, and the average density was substantially lower than in other parts of the myosin map; these features of the SH3 domain were presumably due to a high level of mobility. Nevertheless, our rigor myosin X map possesses a low-resolution feature consistent with the presence of a relatively mobile SH3 domain whose average position/orientation with respect to the motor domain coincides with that seen in the the pre-powerstroke myosin X X-ray structure presented here (Fig. 3a,c,e). Remarkably, modelling the SH3 domain to match the geometry found in the pre-powerstroke structure shifts the SH3 domain out of the way of the rigor conformation of the myosin X lever arm, in comparison with the SH3 domain of myosin V (Fig. 3i,j; Supplementary Movie 1). Thus, our cryo-EM data indicate that the architecture of the converter and the SH3 domain in myosin X are specifically suited to extend the power stroke when entering rigor, compared with structures of myosin V and other myosins[41].

We modelled the myosin X powerstroke by fitting the actin-binding components of the pre-powerstroke structure (upper and lower 50 kDa subdomains) into our rigor cryo-EM map. Comparison of the resulting pre-powerstroke and rigor conformations of the lever arm indicated a net swing of ∼120°. This magnitude is substantially greater than is inferred from our X-ray and cryo-EM[40,41] models of the myosin V stroke (70°) (Fig. 4). Different specific interactions of the converter with the motor domain in both pre-powerstroke and Rigor would thus result in increasing the angle of the stroke by ∼50° for the myosin X motor.

**Model for the step size of myosin X.** By combining the structural data for the motor domain and proximal lever arm with the structure of the distal lever arm and anti-parallel coiled-coil, we constructed the model of the myosin X dimer shown in Fig. 5a (see Methods). This model is consistent with EM data on native HMM (1–954; ref. 13), which report 60–68 nm between heads when the molecule is flattened on the EM grid (Fig. 5b). This model demonstrates that myosin X dimers can step 36 nm on a single actin filament and 57 nm is possible if on an actin bundle. In fact, 52 nm would be a preferred step size on bundles due to the hot spots available on the bundled F-actin surface (Fig. 5c). In contrast this step size cannot be easily accomplished on a single

**Figure 3 | Pre-powerstroke and rigor states of myosin X.** (**a**) Overview of the myosin X pre-powerstroke state (motor domain, blue; lever arm (green, pink, yellow); N-terminal SH3 domain (green). (**b**) Comparison of pre-powerstroke converter positions. (**c**) Comparison of the SH3 domain of myosin V (green) and MyoVc (magenta); orientation as in (**a–e**). (**d**) Interactions of the converter (green) with the N-terminal subdomain (blue) stabilize the specific orientation of the myosin X pre-powerstroke converter. The relay (yellow) and the SH1 helix (red) also specify the converter position. (**e**) Comparison of the myosin X pre-powerstroke state with that of myosin Vc (two molecules in the asymmetric unit are shown in magenta). In (**b**) the converter position closest to that of myosin X is shown. (**f**) Cryo-EM reconstruction of the rigor myosin X motor truncated after the converter, bound to actin filaments reveals a novel position of the converter that places the lever arm helix nearly parallel to the actin helical axis. The density map and corresponding molecular model are rendered in slate blue for the core of the myosin motor, green for the converter and grey for actin. A green rod indicates the extrapolated position of the lever helix, based on the orientation of the converter. The modelled coordinates of the N-terminal SH3 domain are depicted in green. An arrow indicates the direction of myosin force production. (**g**) Close-up of the rigor density for the myosin X converter, with coordinates from crystallized rigor-like myosin V (pink ribbons) superimposed. (**h**) Similar to (**g**) but with a fitted model of the myosin X converter, rendered in green; a rotation of ∼30° with respect to myosin V is predicted from this fit. See also Supplementary Fig. 9 and Supplementary Movies 1 and 2. (**i**) View of the fitted coordinates as seen from the opposite face of myosin as panels (**f–h**); this view reveals that the predicted position of the lever arm helix in rigor is incompatible with the conformation of the SH3 domain found in rigor myosin V (magenta). Steric clashes of main-chain atoms are indicated by coloured spheres; (Supplementary Movie 1). (**j**) Low-pass-filtered rendering of the rigor myosin X density map is consistent with movement of the SH3 domain away from the lever arm helix in order to accommodate the new lever position. Modelled position of the SH3 domain in myosin X is identical to that in (**e**).

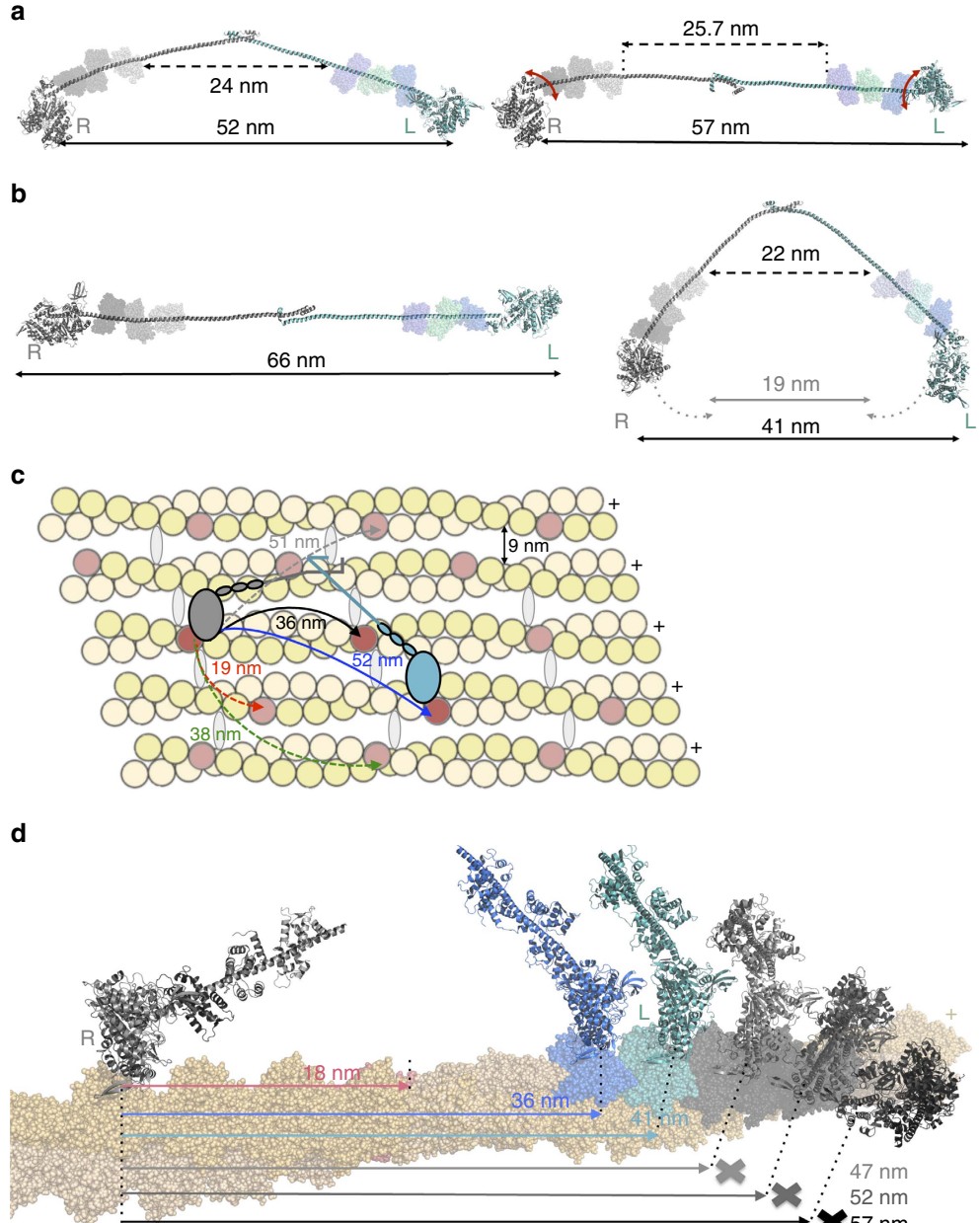

**Figure 5 | How myosin X makes up to 55 nm step.** (**a**) Model of a two-headed myosin X using the structures solved here (SAH-CC-SAH of 24 nm as found in the crystal structure, or ∼25.7 nm, the average found in MD simulations). (**b**) Left: when the heads are flatten, the molecule can reach up to 66 nm as observed for myosin X in EM grids[13]. Right: bending of the lever arm in particular in the SAH allows to explore different spacing between heads and thus find different hot spots. While 52–57 nm is favoured (molecule flat and straight) 40 nm is also possible. Twenty nanometres requires more bending and is thus less favourable. (**c**) Model of myosin X (rear head grey) stepping on an actin bundle. The parameters for the actin bundle were derived from ref. 45. The hot spots are marked with red. Note that 52 nm is favoured on the next filament, while 38 nm would occur if the lead head would step by exploring the bundle. Note that 52–60 nm is not possible on the filament in which the rear head is bound. (**d**) Hot spots for myosin binding on a single F-actin filament. A model of MD-3IQ is modelled as the rear head (grey) and the lead head (colour). Note that the helicity of F-actin is not favourable for the lead head attachment at 47–57 nm from the rear head, while 36–41 nm steps are possible.

actin filament due to the helical repeat placing the actin monomer at 52–57 nm distance on the opposite side of the filament (Fig. 5d), and thus inaccessible to the lead head. However, the flexibility within myosin X (as shown in Fig. 5b right) allows its heads also to explore shorter distances. The lead head can thus reattach at the actin sites 36–46 nm from the rear head (Fig. 5d). Note, however, that distance of ∼19 nm between the heads would require bending or melting of the helical region adjacent to the dimerization region (Fig. 5b right).

## Discussion

In the present study, we demonstrate that the proper dimerization of human myosin X involves formation of an anti-parallel coiled-coil, as was previously suggested[15] but not achieved in earlier single-molecule experiments. We also describe the lever arm of this myosin in the context of a native dimer and show that the distal SAH (846–884) is not involved in dimerization despite the presence of hydrophobic residues and we show that residues distal to 884 do not form an SAH in this structure as previously

reported for a monomer (884–909; ref. 14). This delimitation between lever arm and dimerization region was not obvious since the sequence of both regions are similar in nature and contain a number of hydrophobic residues. The structure of the 'hinge' of the motor that connects the two heads of a dimer is thus described for the first time, as well as the length and structure of the lever arm and the dimerization region of myosin X in its native context.

We found that full-length myosin X, as well as a zippered dimer designed to maintain the anti-parallel coiled-coil, show enhanced motility on fascin-bundles of F-actin filaments. The velocity that we observe on actin bundles closely matches what is observed in filopodia in cells[32], but is more than twice as fast as any previously published values for zippered dimers[27–31]. It is also nearly twice as fast as we observe on single actin filaments. The velocity on bundles is faster than would be expected from the step-size distribution, which suggested that the rate-limiting step of the ATPase activity on actin bundles is accelerated. We previously observed an acceleration of ADP release when we greatly truncated the lever arm of myosin V, forcing it to step off-axis on a single filament[43]. In the case of myosin X, we demonstrate that the rate-liming step that is accelerated on bundles is the transition that is monitored by a change in pyrene-actin fluorescence, which is the step that immediately precedes the ADP releasing transition[37]. Since this step is only accelerated for the myosin X dimer (HMM) and not for the monomer (S1) on a bundle, the basis of the acceleration must be off-axis strain that is generated when the lead head attaches to a different actin filament from the one to which the rear head is bound.

Some earlier reports[27,28], but not all[29], had proposed that myosin X is selective for actin bundles. But these reports measured a myosin X step size of ∼18 nm, based on a chimeric construct, while we now report that full length myosin X has much larger step sizes on either filaments or bundles. While both the zippered HMM and full-length dimer move processively on single and on fascin-bundled F-actin filaments, there was more robust movement on bundles, with longer run lengths and faster velocities than observed on single actin filaments.

As described in Supplementary Note, our results allow us to explain why different artificial myosin X dimers result in significantly different motility properties. These previous results with myosin X chimeras[27–31,44] illustrate the point that the preferred step size is the primary factor in dictating whether a myosin is better suited to stepping on single actin filaments or on actin bundles in vitro. If the architecture of the motor imposes a preferred step size that cannot easily be achieved on a single actin filament, an actin bundle becomes the preferred track. This is the same mechanism that must (at least in part) allow the native dimerization of myosin X to drive preferential trafficking on actin bundles in cells. The most accessible actin site on the same filament is ∼36 nm away, with sites at 41 nm away also accessible (Fig. 5d). However, sites as distant as 47–57 nm are not easily accessible due to the actin filament twist. In contrast, EM and AFM images of fascin bundles[45] indicate that the most accessible sites on the neighbouring filaments in the bundle are at ∼52, 57, 38 and 19 nm away (Supplementary Fig. 10), in good agreement with the step-size distribution we measure on fascin-actin bundles. All of these steps are explored within the step-size distribution, with peaks at these distances fitted by Gaussians (Fig. 2). In the case of our full-length or zippered myosin X dimers, the geometry of the lever arms and the anti-parallel coiled-coil leads to a preferred distance between heads of either 52 or 40 nm, and possibly 57 nm, all of which are well suited to stepping on bundles. A distance of 52–57 nm is not suited for the actin repeat geometry of a single filament (Fig. 5d), and thus

myosin X uses the flexibility in its lever arm to accommodate and shorten the distance between heads (Fig. 5b), to step preferentially at 36 nm if constrained to interact with a single filament. Although target zones on actin bundles exist for 19 nm steps (Supplementary Fig. 10), and some steps are observed at this distance, shortening the head–head distance of myosin X to 19 nm is disfavoured due to the presence of the anti-parallel coiled-coil that biases the molecule towards flat anti-parallel orientations of the lever arms that leads to greater separation of the heads (Fig. 5a,b). This likely explains why less frequent stepping is observed at 19 nm on bundles.

Several components of the myosin X architecture contribute to achieve large steps. We report here that myosin X has a larger stroke than myosin V (120° rather than 70° (ref. 41)). This angle is likely an underestimate of the real angle between the two lever arms. Compliance in the lever arm, in particular at the pliant region (located at the end of the converter, before the first IQ/CaM motif of the lever arm[46] (Supplementary Fig. 11)), can allow the heads to separate further (Supplementary Fig. 12a). This occurs in fact also for myosin V since its 25 nm stroke[33] differs from the 36 nm step size measured (Supplementary Fig. 12b). Thus, compliance can allow the lever arms to adopt a flatter configuration (Fig. 5a, left) to reach 57 nm steps on bundles. Our MD2IQ pre-powerstoke structure indicates this could occur without steric hindrance between the CaM and the rest of the motor (Supplementary Fig. 11). The fact that the distal part of the myosin X lever arm is more flexible is likely important as well to cover exploration of a larger surface and favour binding of the lead head on a target actin-binding zone. Note, however, that in lever arms containing components with considerable compliance, such as myosin X or myosin VI, it is likely critical to have a relatively stiff, IQ-containing segment immediately proximal to the motor in order to bias the attachment of the lead head both in the preferred direction of movement, as well as toward the optimal actin target sites. Note also that the anti-parallel coiled-coil adds not only to the distance between heads but also favours a flattened configuration of the lever arms, biasing the heads toward the most distant target zones. Myosin X thus adopts a relatively 'flat' geometry (that is, lever arms stay close to the actin filaments) along a bundle in order to reach 52–57 nm (Fig. 5c, Supplementary Fig. 12a), while stepping 36–38 nm on a single filament or bundle would involve a geometry in which the lever arms would be more 'upright', or more perpendicular to the actin filament, (as seen for myosin V). It could be that this geometry is disfavoured in the crowded cellular environment, further biasing towards the longer steps for class X myosins.

While the earlier model of Lu et al.[15] did incorporate an anti-parallel coiled coil, as does our model, a major difference is that the model of Lu et al. did not account for the largest steps that we observe with native full-length myosin X, since their model could not predict the greater lever arm swing of our model. Both myosin VI and myosin X contain SAH domains, but in the case of myosin X, it is clear that the SAH domain is part of the lever arm[13,14]. This does not appear to be the case for the myosin VI dimer since its removal doesnot impair myosin function in endocytosis and golgi morphology[22] (Supplementary Fig. 12c).

Myosin X facilitates initiation and elongation of filopodia, which implies favouring formation of parallel bundled F-actin filaments. Myosin X is unique in this role and likely has different roles in this process. Interestingly, initiation of filopodia formation has been described upon over-expression of the tail-less myosin X HMM in cells deficient in myosin X[4]. The special structure and flexibility of myosin X lever arms and dimerization region could favour assembly of F-actin into parallel bundles when the motor reaches out for two different filaments and brings

them together. The myosin X stepping mechanism allows processive and fast transport on the actin bundles of the filopodia, as compared with stepping on single actin filaments, which likely contributes to filopodia elongation. This fast transport on bundles is likely critical, considering that actin undergoes retrograde flow in the filopodia and that neurons can extend filopodia at 500 nm s$^{-1}$ (ref. 47). Once at the tip of filopodia, myosin X is constrained by physical barriers, including the binding of its tail to integrins and the cell membrane. Being part of the tip complex, myosin X likely associate and dissociate with the bundled actin via one or both heads but its specific role in this complex is unknown.

Thus in conclusion, the myosin X structure is optimized for movement on actin bundles, which is critical for its role as a fast filopodial transporter, and perhaps for its role as an actin organizer to promote filopodia formation[4]. Myosin X achieves this by having a preferred stepping behaviour that is best suited for actin bundles via a combination of its anti-parallel coiled-coil, long and flexible lever arms, altered powerstroke and accelerated kinetics.

## Methods

**Myosin X constructs, expression and purification.** Full-length human myosin X (*Myo10*, amino acids 1–2,058) was constructed with an N-terminal Flag tag fused with mWasabi (PCRed out of PmWasabi-NT, Allele Biotechnology) and a three amino-acids (GGR) spacer. We did not want to attach any probes to the C terminus of myosin X in order to avoid potential impact of the placement of the labels on the behaviour of the full-length myosin X molecules. We chose the fluorescent protein mWasabi as the N-terminal fusion protein, as opposed to other forms of GFP, to eliminate any weak dimerization from the GFP, since mWasabi has been engineered to not dimerize. Not only is mWasabi monomeric, but it also is 1.6 times brighter than enhanced green fluorescent protein (EGFP)[48]. A Flag tag was added to the N terminus to allow anti-Flag affinity column purification. Both heads were thus labelled, and we tracked the centroid resulting from the positions of each of the heads, which could not be individually resolved.

Truncated human Myo10 HMM was constructed from amino acids 1–938, lacking PEST, PH, MyTH4 and FERM domains, but including putative dimerization domain. Following amino acid 938, a 19 amino-acid linker (SEGGSGGSGGSGGSAASAA) was added, which was followed by a Leucine zipper, an AVI tag (GLNDIFEAQKIEWHE) and Flag tag at the C terminus. The Human myosin X MD-3IQ-SAH was constructed by truncating at Gln 851 with a C-terminal Flag tag (encoding DYKDDDDK) while the Human Myo10 MD and MD-2IQ constructs were truncated at Glu 741 or at Ile 793, respectively, with a C-terminal Flag tag (encoding DYKDDDDK). All these Myo10 constructs were produced using co-expression with CaM in the baculovirus expression system, as previously described[36]. Actin-activated ATPase assays were performed as previously described[36].

The Bos Taurus Myo10 IQ3-SAH-CC construct from residues L786 to D933 with N-terminal 6-His Tag was subcloned into an expression vector derived from pProEx-HTB. IQ3-SAH-CC co-expressed in BL21 (DE3) cells with CaM subcloned into a pNew vector were grown at 37 °C and induced with 0.5 mM isopropyl 1-thio-β-d-galactopyranoside for 16 h at 20 °C. Cell pellets were resuspended in 20 mM Tris, pH 7.5, 50 mM KCl, 40 mM imidazole, 0.5 mM tris(2-carboxyethyl) phosphine (TCEP). After lysis and clarification, IQ3-SAH-CC + CaM were co-purified by nickel affinity chromatography (GE Healthcare). The column was washed, and the sample was eluted with a buffer containing 300 mM imidazole. The IQ3-SAH-CC + CaM complex were further purified on a HiLoad Superdex 16/600 200 prep grade column (GE Healthcare) equilibrated with buffer 10 mM Tris, pH 7.5, 50 mM KCl, 1 mM TCEP. Production of IQ3-SAH-CC + CaM labelled with seleno-methionine was performed with the same protocol, except that for the cellular culture we used the SeleMet medium from Molecular Dimensions Ltd.

Recombinant DNA of human myosin Vc was generated to express a truncated myosin construct containing the motor domain of this myosin using the baculovirus expression system. A C-terminal truncation was made corresponding to D754 (MD) with a Flag tag appended via a glycine to the N terminus to facilitate purification.

**Labelling of myosin X constructs.** Full-length human myosin X was labelled with an N-terminal mWasabi green fluorescence protein while myosin X HMM (zippered dimer was labelled with quantum dots QDOT525. The AVI tag of Myo10 HMM was first biotinylated by a biotin ligase (BirA; Avidity LLC) according to the manufacture-recommended protocol, then conjugated with QDOT525 streptavidin (Invitrogen) by incubating in 4:1 ratio (QDot:protein) at room temperature for 10 min.

**Actin filaments and fascin-bundled actin filaments.** Actin was purified from rabbit skeletal muscle as described previously[36]. Fascin (gift from Dr Steve Almo) was expressed in *E. Coli* and purified according to a published procedure[49]. Rhodamine-phalloidin-labelled F-actin and fascin-bundled actin filaments were prepared as previously described[29]. Single F-actin filaments were prepared either with or without 10% biotin-labelled G-actin. The bundled F-actin were prepared by mixing and incubating 3 μM fascin and 8 μM F-actin on ice and stored at 4 °C for 2 days before use.

**Kinetic experiments.** Actin-activated ATPase assays and transient kinetic assays were performed as previously described[36]. The exception was that we additionally performed kinetic experiments that used actin bundles, generated as previously described[29], for some experiments in place of F-actin. For the measurement of $P_i$ release, $P_i$ that is released from myosin is detected by binding to a phosphate binding protein that is labelled so that a change in fluorescence is observed upon $P_i$ binding. Turnover is inhibited after the initial transient by including MgADP in the final mix with 0–50 μM actin. The rate of quenching a pyrene label on actin following rapid mixing of MgATP with myosin, followed by a second mix with pyrene-labelled actin (0–50 μM) that also includes a high (MgADP) to stop further cycling. The rate of ADP release from the actin-myosin complex was measured by binding mantADP to the myosin, then competing it off with unlabelled ADP and measuring the rate of fluorescence decrease as the mant signal was quenched by exposure to solvent.

**Single-molecule motility assays.** A FIONA[50] type of assay was used for single step-size measurements. Both step-size and run-length assays were conducted as previously described[50]. For full length, mWasabi myosin X, 10% biotin-labelled single and bundled F-actin filaments were used. In case of single F-actin filaments assay, 5 μM F-actin stock diluted 250 times in assay buffer (50 mM KCl, 2 mM MgCl$_2$, 10 mM imidazole, pH 7.0, 1 mM EGTA, 1 mM dithiothreitol (DTT)), while for bundled F-actin filaments assay, 8 μM fascin-bundled F-actin stock diluted 50 times in assay buffer, was introduced into a flow chamber where filaments bound to the film surface through streptavidin–biotin–BSA complexes. For QDot525 labelled Myo10 HMM, non-biotin-labelled single and bundled F-actin filaments were used. The filaments were diluted and introduced in the same way but bound to the film surface via fascin instead. Unbound filaments were washed out with assay buffer. The assays were initiated by either (1) diluting full length myosin X at a high concentration (∼10 μM) to a concentration of 7 nM into a motility assay buffer immediately before flowing into the microscope chamber, or (2) flowing the myosin X HMM zippered construct (7 nM) into the chamber in the same motility assay buffer. Approach #1 was used because we observed that some full-length mWasabi myosin X construct dimerization occurs without the use of actin-clustering[26], if it was maintained at a high concentration and diluted just before introduction into the chamber. This is consistent with the earlier EM results[13] that suggested that myosin X dimerizes with ∼μM affinity, and with Lu *et al.*[15], who reported a Kd of 0.6 μM for the anti-parallel coiled-coil. The motility buffer was: 50 mM KCl, 2 mM MgCl$_2$, 10 mM imidazole, pH 7.0, 1 mM EGTA, 1 mM DTT, 0.04 mg ml$^{-1}$ BSA, 0.5 μM CaM and 2 μM or 1 mM Na$_2$ATP) supplemented with an oxygen scavenging system (5 mg ml$^{-1}$ glucose, five units glucose oxidase, 0.5 mg ml$^{-1}$ catalase). The motility was imaged by using a Multicolour Leica AM TIRF MC System. The motility was imaged by using a Multicolour Leica AM TIRF MC System. A high-sensitivity and high-speed electron multiplying charge coupled device (EM-CCD) camera (ImagEM-CCD Camera C9100-13; Hamamatsu Corporation) was used with the system for image acquisition. The assay was carried out at 30 °C.

To analyse the data, individual trajectories were extracted from image sequences by means of ImageJ (NIH, Bethesda, MD) and ImageJ plugins: ParticleTracker and SpotTracker[51]. Image sequences were pre-filtered with a LoG filter (MexicanHat) as described[52] to improve signal-to-noise ratio before tracking trajectories. A step-fitting algorithm[52] implemented in MatLab (The MathWorks) was used for step-finding and step-size measurements. The mean run length was determined by averaging pooled data. The run length distribution was also analysed by nonlinear least-squares fitting of cumulative distribution from $X_0$ to infinity to $1 - \exp[(X_0 - X)/\lambda]$ as described by Thorn *et al.*[53]. The data obtained was fitted by applying the normalMixEM procedure in the mixtools package in R[54]. The mixtools package is one of several available in R to fit mixture distributions or to solve model-based clustering. The procedure was applied with its default values to the step-finding data obtained by the step-finding algorithms[51]. To minimize under or over-fitting the mixture distribution, the number of components to fit each data set was selected by cross-validation (Supplementary Fig. 5) of corresponding data set.

**Crystallization and X-ray data collection.** Crystals of Myo10 MD, Myo10 MD-2IQ, Myo10 IQ3-SAH-CC and myosin Vc MD in the pre-powerstroke state were obtained using the hanging-drop vapour-diffusion method. Spontaneous nucleation of myosin X motors occurred at 277 K and at 290 K for Myo10 IQ3-SAH-CC with equal amounts of reservoir solution and stock solution of the protein. The best crystals were obtained using seeding approaches. Before freezing, crystals were transferred stepwise into a final stabilization solution containing ethylene glycol or glycerol as cryo-protectant. X-ray data sets were collected at

100 K at the SOLEIL Proxima-1 beamline. The Myo10 MD protein at 12.5 mg ml$^{-1}$ in 10 mM HEPES pH 7.5, 50 mM NaCl, 1 mM DTT, 1 mM NaN$_3$ supplemented with 2 mM Mg$^{2+}$ADP and 2 mM beryllium fluoride crystallized under condition 10% polyethylene glycol (PEG) 8000, 50 mM Tris pH 7.5, 1 mM TCEP and 125 mM lithium sulfate. The crystallization condition for Myo10 MD-2IQ construct incubated with 2 mM Mg$^{2+}$ADP and 2 mM vanadate was 7.5% PEG 10000, 100 mM MOPS pH 7.0, 1 mM TCEP and 50 mM magnesium acetate. Myo10 MD and MD-2IQ belonged to P1 and P2$_1$2$_1$2$_1$ and diffracted up to 1.8 and 3.1 Å, respectively, (Supplementary Table 2). Spontaneous nucleation of myosin Vc occurred with reservoir solution containing 10% PEG8k, 50 mM MES, pH 6.5, 1% dimethyl sulfoxide and stock solution of the protein at 13 mg ml$^{-1}$ in 10 mM HEPES pH 7.5, 80 mM NaCl, 1 mM DTT, 1 mM NaN$_3$, 2 mM Mg$^{2+}$ADP and 2 mM vanadate. Crystals of Myo10 IQ3-SAH-CC at 12 mg ml$^{-1}$ were obtained with reservoir solution containing 14% PEG 3350, 100 mM ammonium citrate pH 7.0, 2% dioxane. All the images of the data set were processed and integrated with the program XDS[55]. A single-wavelength anomalous diffraction data set of a seleno-methionine substituted IQ3-SAH-CC was collected and phased using the SHELX program suite. The seleno-methionine IQ3-SAH-CC was used to solve the native data set diffracting up to 3.5 Å in P4$_3$2$_1$2$_1$ space group by molecular replacement. The crystals of the motor domains were solved by molecular replacement using the program Phaser[56]. Refinement was performed using Coot[57] and BUSTER or PHENIX[58]. The final statistics are presented in Supplementary Table 2 and the electron density map in Supplementary Fig. 13.

**Molecular dynamics simulations of the SAH dimer.** We modelled the IQ3-SAH-CC/CaM complex by adding two IQ3 regions with CaM bound[59] to the crystal structure of the SAH-CC dimer (residues 812–926). This model was used to perform molecular dynamics simulations with an implicit-solvent model. The simulations use the CHARMM22 all-atoms force field[60] along with the fast analytical continuous treatment of solvation (FACTS) implicit solvent model[61], as implemented in the CHARMM program (version c38b1). FACTS parameters were Tfps = 3, dielectric constant = 1.0, $\kappa = 4.0$ and $\gamma = 0.015$. An implicit solvent was preferred over explicit solvation because both the size and the extended shape of the SAH dimer would have made the MD simulations unreasonably expensive. Moreover, the use of the FACTS implicit solvent model in combination with the CHARMM force-field had been previously validated for the study of SAH sequences by Wolny et al.[14]. N and C termini of each chain were, respectively, blocked by acetyl and N-methylamide groups. The final model includes four chains and 9,659 atoms in total. This model was energy-minimized using 1,000 steps of the steepest-descent algorithm followed by 2,000 steps of the adopted-basis Newton-Raphson algorithm. Following energy minimization, the structure was gently heated up to a temperature of 300 K using successive, 100 ps-long molecular dynamics runs at $T = 50, 100, 150, 200$ and 300 K. Harmonic restraints with a force constant of 10 kcal mol$^{-1}$ per Å$^2$ were applied on the backbone atoms during the heating. A 1 ns equilibration was then performed, under harmonic restraints with a force constant of 5 kcal mol$^{-1}$ per Å$^2$. We then carried out a 100 ns-long production simulation with no restraints. Temperature control was achieved using Langevin dynamics with a friction coefficient of 10 ps$^{-1}$ (heating and equilibration runs) or 1 ps$^{-1}$ (production run). The rigid-body translation and rotation of the protein were cancelled every 500 steps to avoid exaggerated drift. Covalent bonds involving hydrogen atoms were constrained using SHAKE, allowing the use of a 2 fs time step. Visualization and analysis of the simulation trajectories were done using VMD 1.9.2[62].

**Electron cryomicroscopy.** Actin was polymerized and exchanged into EM buffer (5 mM MOPS (pH 7), 5 mM KCl, 1 mM MgCl$_2$, 1 mM TCEP), to a final concentration of ∼200 μM. Actin was then diluted 10-fold into EM buffer and mixed with myosin X motor domain (50 μM, in EM buffer + 50 μM ADP) to a final concentration of 14 μM for actin and myosin (1:1 stoichiometry). Apyrase was added to a final concentration of 0.02 units ml$^{-1}$, and the resulting mixture was incubated on ice for 15 min. Subsequently, 2 mM EDTA was added for another 1 min incubation period. For vitrification, 3.5 μl of the final mixture was applied to a glow-discharged holey carbon grid (Quantifoil), incubated for 1–2 min in a humidified chamber, then blotted for 2 s and plunge-frozen into liquid ethane using an FEI Vitrobot Mark II automated system. Samples were imaged at 27,000 × magnification in an FEI-F20 electron microscope equipped with a side-entry cryo-holder, using a Gatan K2 Summit direct electron-counting camera operating in single-counting movie mode (image size was 3,710 × 3,838 pixels, pixel size was 1.867 Å). A total of 50–56 frames were obtained for each sample area, with a frame collection rate of 3 per second and a net dose over all frames of ∼45 electrons per square Å of sample area.

**Refinement and reconstruction of cryo-EM data.** Movie frames were aligned and summed using the *dosef_driftcorr* motion correction software package[63], with minor modifications to permit CPU (central processing unit)-only computations in the absence of a suitable graphics card. Defocuses and astigmatism parameters for the micrographs were estimated using the CTFFIND3 program[64], and were subsequently used to phase-flip the micrographs before initial extraction of filament images. Image processing and three-dimensional reconstruction

methods closely follow the protocol described in (Kang et al.)[65]. Briefly, initial refinements were performed using the SPARX implementation of the iterative helical real space reconstruction (IHRSR) method[66], using a low-pass filtered map of the actomyosin complex as an initial reference (40 Å resolution). Using the obtained alignment parameters, discontinuities in the filaments were identified and discarded, and the alignment parameters were then mapped back onto the micrographs in order to re-box the filaments with subunit-level precision. Four cycles of FREALIGN refinement were then performed, using specialized helical adaptations[67]; the refinement target option of each round of refinement was successively reduced in the sequence 30, 25, 20 and 15 Å. The final reconstruction utilized 18,949 particles, where a box was extracted from the micrographs separately for each individual subunit identified, and no symmetry was applied during three-dimensional reconstruction. The resolution of the resulting reconstructions was estimated by a gold-standard procedure in which the filaments were separated into two groups, (9,346 and 9,603 particles, respectively), and all SPARX and FREALIGN refinement steps were performed separately on both groups. Soft-edged masks were applied to the gold-standard volumes before Fourier shell correlation (FSC) computation, in order to minimize effects due to noise in the solvent region of the map, and also to exclude peripheral ends of the filament (near the volume boundary) where the map resolution was attenuated. The masks were thus truncated in the axial direction of the filament in order to select only the central three subunits in the reconstructed volumes yielding estimates of 9 Å for the resolution of actomyosin, and slightly higher for actin alone (0.143 criterion) (Supplementary Fig. 9a). In order to assess the resolution of the converter, a soft-edged mask was generated from a low-pass-filtered rendering (30 Å) of the fitted converter coordinates. A B-factor of −600 was applied to the maps before rendering for display and fitting calculations.

**Molecular docking and model generation with cryo-EM data.** A homology model of myosin X MD in the rigor state was generated by SwissPDBViewer (http://spdbv.vital-it.ch/) using the myosin Va rigor structure (1OE9; ref. 40) for a template, but replacing the N-terminal SH3 domain by that of myosin X (from the X-ray 1.8 Å resolution pre-powerstroke MD structure described here). Indeed this myosin X pre-powerstroke structure has revealed that this domain is quite different from that of myosin V (Fig. 3c). Atomic coordinates of actin (3j8a; ref. 68) and Myo10 MD were fit into density maps using local refinements based on cross-correlation with synthetic models X-ray crystal structures rendered at 9 Å resolution (backbone atoms only), via the 'Fit in Map' function of the Chimera package[69] from the Computer Graphics Laboratory, University of California, San Francisco.

For fitting of the converter domain into our cryo-EM maps, we switched to a more comprehensive rigid-body search method to assess the robustness of the resulting solutions. From each independently-refined half-data set reconstruction as well as the one derived from all the data, we excised a small volume (32 × 32 × 32) containing the converter domain, and then used the Situs package[70] to perform an exhaustive six-dimensional search using our coordinates of the myosin X pre-powerstroke converter (residues 674–738) as a search model. For search parameters, we employed 5° angular increments and activated the laplacian filter option; 9 and 10 Å resolution cutoffs were applied, respectively, for the pair of half-data refinements as well as the full data set one. The resulting estimates of the converter position/orientation differ from one other; the extent of the differences reflects residual noise present in the reconstructions (noting that the residual noise in each half-data set refinement is essentially independent of the other half-data set refinement). For each of the two half-data set refinements, the top-scoring solution thus obtained falls within a 10° cone angle of the other, defining an orientation of the converter that is nearly parallel to the axis of the actin filament and offset 28°–38° from the orientation seen in the rigor-like myosin V X-ray structure (Fig. 3f–h; Supplementary Fig. 9b–d; Supplementary Movie 2). The three alpha helices in the resulting alignments neatly coincide with well-defined, elongated features in the full-data set density map (Fig. 3g,h; Supplementary Movie 2). This analysis thus indicates that the cryo-EM data uniquely define the orientation of the converter domain with respect to the rest of the myosin motor domain.

**Building of the molecular model of myosin X.** To build an atomic model of dimeric myosin X, the converter and two IQ motifs of MD-2IQ in the pre-powerstroke state were superimposed on the converter of the rigor model from the cryo-EM data to generate the rigor MD-2IQ coordinates. The third IQ was modelled using the coordinates of the 2IQ$_{1-2}$/CaM myosin V complex (2IX7 (ref. 60)) on both the pre-powerstroke and rigor heads. The structure for the missing residues between these myosin X MD3IQ rigor and pre-powerstroke constructs was obtained using the coordinates of our IQ3-SAH-CC structure solved in this study after relaxation dynamics by molecular dynamics (see Fig. 5a right).

All molecular graphics images were rendered by UCSF Chimera or were generated with Pymol (http://www.pymol.org/).

**Data availability.** The atomic coordinates and structure factors have been deposited in the Protein Data Bank, www.pdb.org, with accession numbers 5I0H (Myo10-MD-PPS), 5I0I (Myo10-MD2IQ-PPS), 5HMO (Myo10-SAH-CC), 5HMP (Myo5c-MD-PPS). Model and maps for the cryo-EM reconstruction of rigor

Myo10 have been deposited in the Protein Data Bank and the EMDB with accession codes 5KG8 and EMD-8244, respectively. The authors declare that all relevant data supporting the findings of this study are available on request.

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

## Acknowledgements

We thank beamline scientists of PX1 (SOLEIL synchrotron) for excellent support during data collection. We gratefully acknowledge the Yale CCMI Cryo-EM facility for support and maintenance of cryo-EM equipment. We thank staff for maintenance of the Yale Cryo-EM facility and High-Performance Computing facility. A.H. was supported by grants from the CNRS, ANR BLAN10 and ANR-13-BSV8-0019-01, Ligue contre le cancer and ARC. H.L.S. was supported by NIH grants DC009100 and HL110869. C.V.S was supported by NIH grant RO1-GM1053001. The AH team is part of Labex CelTisPhyBio 11-LBX-0038 and IDEX PSL (ANR-10-IDEX-0001-02-PSL).

## Author contributions

V.R., Z.Y. and T.I. contributed equally to this work. V.R. and T.I. determined X-ray structures and analysed them with A.H. Cryo-EM structures were determined by K.Z. and C.V.S and analysed by C.V.S. Single-molecule studies were performed by Z.Y. and E.D.Y. and were analysed by Z.Y. and H.L.S. Molecular dynamic simulations were performed by F.B. and M.C. Biochemical experiments were performed by V.R., X.L., P.H., F.S. and B.A. T.Z. performed kinetic studies. A.H. and H.L.S. conceived the project, oversaw the experiments, analysed them and wrote the manuscript with help from V.R. and C.V.S.

## Additional information

**Competing financial interests:** The authors declare no competing financial interests.

