## [Peer Review File · Nature Communications]

Reviewer #1 (Remarks to the Author)

This is an interesting paper that shows the first structure for the motor domain of myosin X in the 'pre-power stroke state', which has an interesting and novel converter domain organization compared to other myosin motors, and how this changes during the power stroke, using CryoEM, and fitting in of the structure a post-powerstroke motor domain bound to actin. The key feature here is that there is a large movement of the converter domain from pre-to post power stroke. In addition, the converter domain positions the lever in a more parallel orientation with respect to the actin filament compared to other myosins, a potential adaptation to the filopodial function/localization. The paper also shows a low resolution structure for the IQ-SAH-CC region of the proteins, and some SAXS analysis for the region just beyond the CC, and PH domains. It also looks at the stepping behavior of dimerized myosin X along actin filaments for HMM and full length constructs along bundled actin and single filaments, although this part of the paper I found less novel, as this has been done many times before (as summarized in supplementary table 1 of this paper).

Comments:

Fig. 1A is incorrect. Both references 12 and 13 show that the SAH domain extends past residue 846 into the entire region that is colored red in this diagram: e.g. we already know that the SAH domain is formed by both the blue and red regions. The 814-846 stretch was characterized by extensive peptide analysis in ref 12, but that reference did also measure the length of the head of myosin X in an HMM construct in both monomers, and in the rarer dimers and this measurement provided an estimate of the length of the SAH domain, showing that it extended through the 'red' and 'blue' regions in fig 1A. In reference 13, a longer sequence (residues (813-909) from myosin X was characterized in vitro and shown to behave as a SAH domain.

Additionally, in the text referring to Fig. 1A, which shows an anti-parallel coiled coil, the original publication showing the presence of an anti-parallel coiled coil by NMR (Ye et al., 2012, PNAS) just distal to the SAH domain ought to be mentioned at this point.

The 'controversy about lever arm designs' discussed in the introduction refers to some papers about myosin VI, but omits a key paper (Spink et al., 2010, NSMB), which did show the 'lever' of myosin VI by SAXS for the first time.

Overall I recommend that the authors replace 'design' with evolution/evolve throughout the paper, to avoid any confusion with 'intelligent design' and the 'creator'.

The introduction refers only to dimerized myosin X, but there is also still some controversy about how/when myosin X might really dimerize, (as well as myosin VI), and this is really glossed over. Isolated molecules of both myosin X and myosin VI have been shown to be monomeric by electron microscopy (Lister et al., 2004, EmboJ) and Umeki et al., 2011, NSMB). This controversy is not really discussed here in the introduction. Thus when the introduction goes on to talk about 'tail-less myosin X dimer' and full-length dimer, the reader does not get to appreciate that there is still discussion in the field about the dimerization status of these proteins. This needs to be addressed.

Results

The authors mention a 'consensus SAH sequence. To my knowledge, there is no consensus SAH sequence. No such sequence has been published anywhere, through analyzing sequences from the large number of proteins that these sequences are found in. The reason for this is that these domains are characterized by being rich in charged residues (E,K and R), but the repeating patterns of these residues in individual domains is highly divergent and thus difficult to align to generate a consensus sequence. Perhaps aligning the sequences from Myosin VI, VII and XVII show some similarities (e.g. as in reference 12), because these SAH domains are all found in the

same type of protein, but across proteins this does not work.

It is important that the authors clarify this in the paper, as while a highly charged proximal region is easy to 'spot' as a SAH domain, it has already been shown that the longer myosin X peptide is additionally a SAH (see above), even though the frequency of charged residues between 846 and 909 is reduced, of course adding to the complexity of trying to identify a consensus SAH sequence. The authors need to change their text to reflect this.

The authors' low resolution structure of the IQ-SAH-CC construct shown in Fig. 1B actually confirms this original published data, rather than discovering an unknown structure. E.g. the statement that '... (E847 - E880) together form a 10.5nm long single alpha-helix and elongates the IQ region' actually does confirm the earlier reports in references 12 and 13. Indeed the statement made by the authors that E487-E880 forms an 'unstructured linker' (from Ye et al., 2012), is misleading, when both refs 12 and 13 show that this region is not unstructured despite not being what the authors term a 'consensus' SAH (but see comments above). The text should be revised to reflect this.

I also found the image presented in Fig. 1B confusing as it shows 814-846 as a solid box, and 846 to 881 as a helix, when presumably the whole region is a helix. It was unclear to me why the entire region was not shown as a helix.

The authors then go on to use molecular dynamic simulations of the SAH-CC-SAH dimeric region. This modelling uses a really helix biased force field (CHARMM22, see e.g. DOI:10.1016/j.bpj.2012.07.042) and runs for just 100 ns. It is therefore not unexpected that the helices do very little, and the modelling adds little to the manuscript. Moreover, this is all done in implicit rather than in explicit solvent. The estimated length of this construct could be achieved from first principles (known rise per residue in an alpha helix), plus the additional length measured from the anti-parallel coiled coil structure without resorting to modelling. I recommend that the modelling is omitted.

The authors then go on to discuss the single molecule motility of human myosin X. It is not clear to me why, despite the presence of an 'anti-parallel coiled coil', the authors still need to dimerize the shorter 'HMM' construct with a leucine zipper (albeit with a longer section of spacer sequence between the anti-CC and the leucine zipper compared to previous studies). And they need to use a separate technique to dimerize full myosin X, by crowding the molecules onto actin filaments in rigor, and then releasing them. Particularly important here is that in previous papers where the authors have used this 'crowding' approach for myosin VI, data was provided on how many molecules were dimerized, and for how long. The same data should be provided here. It would also be highly informative to see images of the 'dimerized' molecules - do they look similar to the rare dimers seen by rotary shadowing EM published in ref 12?

The need to dimerize myosin X in this way reinforces the issue that myosin X only appears to dimerize fairly weakly. It is still not clear (despite quite a few papers on this subject) if it dimerizes in the cell, and if so, by how much? With only ~1000 molecules in the cell, can it ever become clustered enough to dimerize, or not?

With such weak dimerization of the zippered HMM construct, can the authors be sure that the molecule is in fact dimerized through the antiparallel coiled coil, and not solely through the leucine zipper? If the helices that form the anti-parallel coiled coil dissociate, then molecules would only be dimerized through the leucine zipper, which in this case is placed further away from the SAH domain, compared to previous studies. Could this account for the longer 57 nm step in HMM compared to the full length construct (52 nm)?

Can the authors show the raw data for the histograms presented in Fig. 2? E.g. superimpose a number of frames from a time series and show the image, or some other approach?

The data on movement on individual filaments and bundles is interesting (Fig. 2), but I'm not convinced that it makes a radical new development compared to previous data already published from the Rock group and others (Table 1). The step-size plots are selectively over-interpreted to fit the step sizes that generated by their model. All the backward steps are placed into a single 'bin' but not the forward steps. Separate peaks are 'seen' in the histograms for the bundled data, but a slightly different/adjusted analysis of the single filament data, could also show multiple step sizes from eyeballing the data. No statistical analysis is provided. No details are given of the errors (e.g. mean +/- S.D.) for these data. For these reasons I recommend that this data is moved to the supplemental material, and careful consideration is given to these criticisms.

Similarly, the assertion that the two headed construct is 'gated' needs more data, rather than just the difference in ATPase rate. I'd like to see the results for MantADP etc (as performed by the Howard White Group). And what would happen to these measurements if bundled actin, rather than single filaments were used?

For me, and for the field, the most interesting part of the results comes in the next section on the structure of the myosin X motor domain, although this section could be written much more clearly. Nter is not defined (for example). The paragraph needs breaking up into two, with a second paragraph starting on 'the pre-powerstroke structure'. And all this data (currently in supplementary figure 4a,b) needs to be in the main paper, and this section really expanded so we can see in detail how the converter region, and the Nter interaction are important in this motor, and different to that of other myosins. I therefore strongly recommend that this data is given much more prominence in the paper, and rescued from its position mainly in supplementary, and that the stepping data is placed in supplementary data as it is less informative and less novel.

Hidden also in this one paragraph/section is 'we also crystallized motor domain of Myosin Vc in this state' to compare the structure of Myosin X with that of Myosin V. But this is surely not the case, as there are multiple structures of Myosin Va already published! It seems to me that Myosin Vc structure has been solved by these authors, and it has been inserted into this paper as a way of 'publishing' the structure more easily! Moreover, there are no details on the construct, in the methods, how it was made etc etc, and the sudden arrival of a structure of Myosin Vc is surprising. I suggest either removing this structure, or more strongly defending its inclusion. This becomes even more confusing, when it's myosin Va that is compared to MyoX in figure 4c!

Next, the authors investigate the rigor state of myosin X on actin (Fig. 4). The legend to Fig 4 states that they see the first 3 turns of lever arm helix, but this is not labelled on the figure - is this a spike of density sticking out of the converter? Super-imposing an atomic ribbon model on the EM density, makes it impossible to see the EM density. The authors should show a panel first without the model. In addition, supplemental figures 5b-f should be moved to the main part of the paper.

The authors also comment that 'the SH3 domain is different' to other myosins, but do not go on and further explain how it is different from that seen in previous structures. They should.

Finally, the authors need to clarify what the the C-terminal residue is of the construct used here in cryo-EM, so that we can tell how long a section of lever (which ought to be the first IQ motif) we should be expecting to see.

In the last section of the results, the authors go on to model the dimer (Fig. 5a). However, the diagrams of actin shown (Fig 5c and S6b) are so wrong that it undermines my confidence that their conclusions are correct. Actin is shown as having two chains of subunits that are axially aligned rather than offset by a half subunit, and which are left-handed helices rather than right handed. These are really basic errors that are out of place in a paper on acto-myosin structure. They must replace these with correct ones, and revise their conclusions if necessary. The

justification for the axial and lateral offsets they have used between adjacent actin filaments in a fascin bundle, which allows them to calculate distances between hot spots is not referenced in either Fig legend, but it should be. I'm also confused by the way that the myosin molecule is drawn, which seems to be more consistent with parallel dimerization, than anti-parallel (e.g. compare fig. 5b - which is clearly antiparallel, with fig. 5c, which is not).

Moreover, in Fig 5d, the X's and vertical dotted lines and arrows, should have equal increments of distance of 5.5 nm (as implied by the integer values in nm put next to them), but they don't. Instead they get closer together at the right hand side, so that the 57nm one isn't pointing to the last motor at all, it's pointing at the preceding one. This is out of place in a paper on structural biology.

Discussion.

The first paragraph returns to the 'dimerization' discussion, and goes back to stating that the 'dimerization region of myosin VI precedes the SAH', but again, the authors should cover the controversy in this field regarding this statement. And this needs to be addressed at later stages in the discussion.

The assertion in the discussion that 'the 34 residues after the consensus SAH' needs to be rewritten, as these residues were already shown to form a SAH (refs 12 and 13) and this new data only confirms these earlier observations. Moreover, as explained earlier, there is no 'consensus SAH' and known SAH domains can contain hydrophobic residues.

The short discussion on selectivity for actin bundles, does at least concede that this selectivity has been shown before. But I'm still not convinced by how novel the data presented here is. And it's not clear if the anti-parallel coiled coil can still form in HMM when the molecule is dimerized by a leucine zipper as no data is shown to support/discount this idea. And no images are shown of the 'clustered' dimers of full length myosin. And lastly, it is completely unclear to me how leucine zippered constructs reported in earlier publications could 'somehow shorten the lever arm'. What would the mechanism be?

I recommend that the discussion places more emphasis on the structure of the motor.

The discussion would benefit from shortening, and more clarity, and discussion of the following:

The stiffness of the SAH domain is probably about 10fold lower than that of the IQ lever (e.g. seem data from Spudich group in PNAS), and one might expect the SAH domain to be more flexible - how would this affect the data interpretation?

The majority of myosin X in a filopodium is actually found at the tips. Do the authors think that the re-modelled converter/lever that they have found in myosin X might be important for linking myosin X to the ends of actin filaments at the filopodial tips?

Methods

Is there a reason why the human constructs are used in some experiments, but the bovine Myosin X was used elsewhere?

Dimerization of myosin 10 by clustering - needs more detail (see above). The authors observed dimerization without actin clustering 'although (sic) in a smaller fraction', but do not give any details (again see comments above). Were these non-clustered dimers ever used in experiments?

Crystallization: no details are given for the MyoVc MD construct until this section. They should be (or MyoVc data removed)

Molecular dynamics

No details given on how the force field was chosen, they should be

Cryo-EM Methods section

No mention of having any myosin in the frozen sample, nor what concentration of actin they used, nor anything about the composition of the solution that was frozen is provided. The solution termed 'EM' buffer has no buffering species in it, which is hopefully wrong as the pH could be just anything!

What construct of myosin- X was used for the EM, what buffer was it in prior to use for EM and what was the composition of the mixture once the S1 had been mixed with the actin before freezing?

The text in Results and the Fig 4 legend suggests that a motor domain alone (i.e. no IQ motifs) was used, but there is no mention of such a construct in the Methods section.

Did they use the whole S1, but the lever was disordered in the EM or what?

Table S2 mentions crystal structures of MD alone (and other things too (including some myosin-5 constructs), but there is no mention of making these in Methods, so what are they - what residues do they encompass?

A proper Methods section that fully describes (and names) every construct used in the whole study should be provided. These names should be used consistently in the rest of the paper so that there is no ambiguity over which data come from which construct. A Table would help. While Fig S1b is helpful, it does not cover everything that appears to have been used. It would be impossible for anyone to reproduce their work from the information they have given.

Overall, in both the main paper, and in supplemental, there are a lot of typos, etc, so it needs a proper edit. E.g. Fig S3 legend says they've used a monomer truncated prior to the coiled coil. Which bit of coiled coil, so what is the C-terminal residue? They should have a name for the construct in Methods, and refer to it by this name in this Fig legend. Further on in the legend they refer to wild type and HMM as having a single V_{max} . Where is the data for wild type or this this a typo? Then later "S1 was and 19.6" - means what??

Other comments:

1st para of results mentions 'gating' but does not explain what this is, or provide any references.

The SAXS data on the CC-PH1-PH2 seems really redundant to me. While it may show that this construct is extended, I don't see that this data, on its own, supports the idea that the antiparallel CC is not influenced by the C-terminal sequence as claimed. I recommend that this data is removed.

If not removed, the SAXS (Supplementary Fig. 2a-d) and modelling data (Supplementary Fig. 2e-g -) should be presented in separate figures.

A recent paper on myosin X in prostate cancer was published in cell reports (doi: 10.1016/j.celrep.2015.11.012) has been omitted from the introduction (1st paragraph).

Reviewer #2 (Remarks to the Author)

Ropars et al. present a concise study of myosin X. Combining X-ray crystallography and cryoEM,

the authors determined structures of the pre-powerstroke and rigor state of the myosin X motor domain and describe an extreme angle of 120° for the powerstroke. They determined a crystal structure of the dimerization domain including the SAH domain. SAXS data of the full-length protein show that it is flexible in solution. Single molecule studies with new constructs including the dimerization domain allow a more accurate measurement than those that have been performed previously by others and show that myosin X has a different stepping behavior on single F-actin filaments and filament bundles.

The authors excellently discuss their results. Their findings are important and should be published.

However, the manuscript has the following shortcomings:

1. In their introduction and in the rest of the manuscript they do not clearly describe what was there before and has been published previously by others. For example, Lu et al. presented a NMR structure of the anti-parallel coiled-coil dimer region of myosin X in 2012. It is not mentioned in the introduction and in the Results part, Ropars et al. present it as if it has been unknown so far. The only new aspect of the crystal structure is that the linker is a helix as proposed previously (see review by Li et al. 2016).
2. The only result from the SAXS data is that myosin X is flexible as full-length protein. This is not surprising. The models based on the SAXS data as presented in Supplementary Fig. 2 are pure fantasy.
3. The pre-powerstroke structure of the motor domain of myosin X is very similar to that of other myosin motor domains (Vc, VI, Ic, smooth muscle myosin). It is nice to have it, but unfortunately it does not reveal new important details.
4. By actual measures, the cryoEM structure is of low resolution although the authors used a direct electron detector for recording their data. This is surprising and makes the proper fitting of the converter domain - a central aspect of the manuscript - questionable.
5. When presenting their model, the authors do not state that others have proposed the model before (see for example Lu et al. 2012).

Taken together, Ropars et al. confirm and extend known data and present a good explanation for a model that has been previously described by others, for example by Lu et al. 2012. Nevertheless, their findings are important and should be published after the named issues have been resolved.

Other specific points:

1. The authors give exact values without deviations for all measured step sizes although the measurements do not allow for this accuracy. Proper deviations should be given.
2. "...reconstruction was ~10 Å..... sufficient to unambiguously define the position and orientation of most alpha-helical elements in the map." A map at around 10 Å is definitely not high enough to unambiguously define the position and orientation of alpha-helices.

3. What is the resolution of the converter domain in the cryoEM map? Is the resolution high enough to unambiguously fit the model in this region. I suspect that a detailed fit of the last helix of the converter domain and determination of the degree of the lever arm (Fig. 4) is most likely not possible.

Supplementary Figures 5b-e including legends are difficult to understand. Maybe the subfigures can be split in two, one subfigure showing the map/simulated map/envelope and one figure of the same field of view of the models. Which PDB was used for the myosin II model? Why 'Myosin II model' and only 'Myosin X' without 'model'? A movie could be used to make their statements clearer.

4. For me it is not really obvious, how the authors used the 'Fit in map' function of Chimera to perform local refinements for creating the whole rigor model of myosin? It should be stated how the pre-powerstroke state model was fitted (rigid-body fitted into the cryoEM map is not sufficient, as the myosin structure differ in 1-670 between rigor and pre-powerstroke state and thereby is not rigid at all). Why have the authors not flexibly fitted their pre-powerstroke state model into their

rigor density?

5. The authors describe a simple and plausible model for different myosin X step sizes. Is there any evidence how fascin binds and builds up the F-actin bundles? The authors assume a flat and organized structure. Is it not more likely that the F-actin-fascin bundles are less flat or less well organized in vivo? The broad distribution of possible step sizes is likely an indicator for that. This aspect should be discussed.

6. It does not make any sense to give values like: 32.72 +/- 14.27. A decimal number is not meaningful at all in this case and 33 +/- 14 would be correct.

7. Measurement deviations of SAXS data are completely missing.

8. Pymol is not cited.

9. The statement that their work 'may also reveal targets for therapeutic interventions to combat metastatic cancers' is a bit overstated and should be removed.

10. Methods are very rudimentary and one would not be able to reproduce their results based on the descriptions.

11. Fig. 1b/c rotation between the two views of the same models should be indicated by an arrow and angle.

12. p. 5 "...rather than partially structured linker as previously described". This statement is not correct. Lu et al. describe this linker as "semi-rigid helical linker". This is what the authors in principal show with their MD simulations (Fig. 1e).

13. What is ~nanometer resolution?

Dear Dr. Legate:

We feel that we have been able to respond to all of the points that the reviewers made in critiquing our submitted manuscript (NCOMMS-16-03730) as outlined below.

Reviewer #1:

*This is an interesting paper that shows the first structure for the motor domain of myosin X in the 'pre-power stroke state', which has an interesting and novel converter domain organization compared to other myosin motors, and how this changes during the power stroke, using CryoEM, and fitting in of the structure a post-powerstroke motor domain bound to actin. **The key feature here is that there is a large movement of the converter domain from pre-to post power stroke.** In addition, the converter domain positions the lever in a more parallel orientation with respect the the actin filament compared to other myosins, **a potential adaptation to the filopodial function/localization.** The paper also shows a low resolution structure for the IQ-SAH-CC region of the proteins, and some SAXS analysis for the region just beyond the CC, and PH domains. It also looks at the stepping behavior of dimerized myosin X along actin filaments for HMM and full length constructs along bundled actin and single filaments, although this part of the paper I found less novel, as this has been done many times before (as summarized in supplementary table 1 of this paper).*

We thank the reviewer for his interest in our story and for his very careful reading and corrections for the manuscript. We must apologize for a number of errors and omissions in the manuscript. These were unfortunately introduced when some previous corrections were lost due to the fact that an older version was reintroduced in the exchange of manuscripts between authors. These have been corrected. We also emphasize that there are at least two novel features of movements of the HMM and full length myosin X constructs on actin bundles (but not on filaments). First, the velocity on the bundles is twice what is seen on filaments, and matches *in vivo* values, Secondly, a population of larger steps are seen on bundles that have ever been observed for any myosin.

Comments:

Fig. 1A is incorrect. Both references 12 and 13 show that the SAH domain extends past residue 846 into the entire region that is colored red in this diagram: e.g. we already know that the SAH domain is formed by both the blue and red regions. The 814-846 stretch was characterized by extensive peptide analysis in ref 12, but that reference did also measure the length of the head of myosin X in an HMM construct in both monomers, and in the rarer dimers and this measurement provided an estimate of the length of the SAH domain, showing that it extended through the 'red' and 'blue' regions in fig 1A. In reference 13, a longer sequence (residues (813-909) from myosin X was characterized in vitro and shown to behave as a SAH domain.

Additionally, in the text referring to Fig. 1A, which shows an anti-parallel coiled coil, the original publication showing the presence of an anti-parallel coiled coil by NMR (Ye et al., 2012, PNAS) just distal to the SAH domain ought to be mentioned as this point.

We agree with the reviewer that the region 846-880 was shown to form a coiled-coil in the context of the study of a monomeric fragment 813-909. Note however from Figure 1d that there is no way to differentiate what the end of the SAH will be and where the dimerization region would start from looking at the sequence. Our intention was to question whether this 846-880 region which contains hydrophobic residues could in part participate or not in the region involved in dimerization by studying a long fragment 813-938 both by X-ray crystallography, SAXS and molecular dynamics. The question was to define precisely what was the nature of the dimerization region in the native context of this region.

Previous single molecule studies performed with myosin X constructs had clearly indicated that different results could be obtained depending on the construct length and addition of leucine zipper. We took this as an indication that the native dimerization sequence was not maintained in some or all of the previously published constructs. The first report from the Rock's lab PNAS 2008, indicated clearly that 1-920 + Leucine zipper could lead to dimerization of the region proximal to 920 in order to restrict the Dmax for the heads at 18-20 nm. Thus in this construct, it is likely that the myosin X sequence that forms a SAH is involved in dimerization in the context of this chimera in order to account for such short steps. Prior to our study, what was known was that 813-909 could act as an SAH in the context of a monomer, 880-938 could form an anti-parallel dimer, but it was important to show the structure of the native molecule, since small peptides do not always recapitulate the same behavior if additional residues from the native sequence are introduced.

We have now revised the text in introduction (p.3) results (p.5) and beginning of discussion (p.11) to make this logic more clear, we cite the previous knowledge about the SAH and dimerization region more precisely in the introduction and we have also revised Fig.1b to show the helices as cylinders as asked by the reviewer. We also now add SAH clearly for this region since our study does indicate that in the context of the dimer this region stays a SAH as previously shown in the context of a monomer.

The 'controversy about lever arm designs' discussed in the introduction refers to some papers about myosin VI, but omits a key paper (Spink et al., 2010, NSMB), which did show the 'lever' of myosin VI by SAXS for the first time.

This reference has been added p.4.

Overall I recommend that the authors replace 'design' with evolution/evolve throughout the paper, to avoid any confusion with 'intelligent design' and the 'creator.'

We had not even considered the possibility that the word design would imply "intelligent design", since the word intelligent does not appear in the manuscript. We still feel that design is the appropriate word in many instances, and could not find a better one. Nonetheless, we have removed it from the title, and tried to remove it in a number of other places. Whether the design came to be via evolution or by the will of God is beyond the scope of this manuscript. We make no mention of evolution, God, or intelligent design.

The introduction refers only to dimerized myosin X, but there is also still some controversy about how/when myosin X might really dimerize, (as well as myosin VI), and this is really glossed over. Isolated molecules of both myosin X and myosin VI have been shown to be monomeric by electron microscopy (Lister et al., 2004, EmboJ) and Umeki et al., 2011, NSMB). This controversy is not really discussed here in the introduction. Thus when the introduction goes on to talk about 'tail-less myosin X dimer' and full-length dimer, the reader does not get to appreciate that there is still discussion in the field about the dimerization status of these proteins. This needs to be addressed.

We have added a paragraph to describe this in introduction (p.3). We would like to point out that Knight and colleagues [Knight PJ, Thirumurugan K, Xu Y, Wang F, Kalverda AP, Stafford WF 3rd, Sellers JR, Peckham M. (2005) The predicted coiled-coil domain of myosin 10 forms a novel elongated domain that lengthens the head. J Biol Chem. 280: 34702-8.] did in fact see myosin X dimers, as well as monomers, in EM, suggesting that myosin X can exist in either form. In fact, the authors stated, "Both rotary-shadowed and negative-stained images show that about 90% of the molecules of this myosin 10 construct are monomeric (Fig. 5). The remainder of the molecules were

dimers, and this low proportion suggests that there could be a weak equilibrium between monomers and dimers, with a binding constant in the micromolar range.” This is consistent with a K_d for dimerization measured as 0.6 μM by Lu et al 2012. This was the basis of our keeping our full-length construct at $>\mu\text{M}$ concentration until diluted into the motility chamber.

Results

The authors mention a 'consensus SAH sequence. To my knowledge, there is no consensus SAH sequence. No such sequence has been published anywhere, through analyzing sequences from the large number of proteins that these sequences are found in. The reason for this is that these domains are characterized by being rich in charged residues (E,K and R), but the repeating patterns of these residues in individual domains is highly divergent and thus difficult to align to generate a consensus sequence. Perhaps aligning the sequences from Myosin VI, VII and XVII show some similarities (e.g. as in reference 12), because these SAH domains are all found in the same type of protein, but across proteins this does not work.

It is important that the authors clarify this in the paper, as while a highly charged proximal region is easy to 'spot' as a SAH domain, it has already been shown that the longer myosin X peptide is additionally a SAH (see above), even though the frequency of charged residues between 846 and 909 is reduced, of course adding to the complexity of trying to identify a consensus SAH sequence. The authors need to change their text to reflect this.

The text has been changed accordingly (p.5). We removed the word consensus and describe more precisely this region. Rather than consensus, we perhaps should have said unambiguous. Our point was that the a region containing only E,K and R residues can only be SAH, but the introduction of hydrophobic residues makes it less obvious. It is not at all obvious from the sequence where the SAH stops and the coiled coil begins.

The authors' low-resolution structure of the IQ-SAH-CC construct shown in Fig. 1B actually confirms this original published data, rather than discovering an unknown structure. E.g. the statement that '... (E847 - E880) together form a 10.5nm long single alpha-helix and elongates the IQ region' actually does confirm the earlier reports in references 12 and 13. Indeed the statement made by the authors that E487-E880 forms an 'unstructured linker' (from Ye et al., 2012), is misleading, when both refs 12 and 13 show that this region is not unstructured despite not being what the authors term a 'consensus' SAH (but see comments above). The text should be revised to reflect this.

I also found the image presented in Fig. 1B confusing as it shows 814-846 as a solid box, and 846 to 881 as a helix, when presumably the whole region is a helix. It was unclear to me why the entire region was not shown as a helix.

As we have described also for the introduction, we have now revised the text to better describe these different SAH regions and why it is important to confirm with a full length dimerization region what is the exact length of the SAH and what is the native sequence for the dimerization region, as well as the dynamics and stability of the whole region in the context of a dimeric structure of the SAH-cc-SAH, now directly described using the X-ray structure (p.5).

The presence of the hydrophobic residues found in the second part of the SAH made it difficult to distinguish what was SAH and what was part of the dimeric region from sequence alone. One could not exclude that although the monomeric SAH would be stable, a larger part than what had been shown to be part of the anti-parallel coiled-coil could also have been recruited to stabilize the dimerization domain. The fact that the 1-920-zipped constructs (of Rock and colleagues) had

shown that some part of the SAH/Dimerization region was likely to interact and restrain the heads so that only 18 nm could be reached, at least in the context created by that construct.

The authors then go on to use molecular dynamic simulations of the SAH-CC-SAH dimeric region. This modelling uses a really helix biased force field (CHARMM22, see e.g. DOI:10.1016/j.bpj.2012.07.042) and runs for just 100 ns. It is therefore not unexpected that the helices do very little, and the modelling adds little to the manuscript. Moreover, this is all done in implicit rather than in explicit solvent. The estimated length of this construct could be achieved from first principles (known rise per residue in an alpha helix), plus the additional length measured from the anti-parallel coiled coil structure without resorting to modelling. I recommend that the modelling is omitted.

Despite the criticism raised, we believe the modeling results provide important additional information to understand the structure and function of myosin X in the context of a dimer. Regarding the choice of the force field, we would like to point out that a very similar simulation setup was used to study the SAH region of myosin X in the recent work by Wolny et al (Wolny et al. Journal of Biological Chemistry 2014). These authors have validated the use of the FACTS implicit solvent model, which exhibited reversible helix formation events for short SAH-like sequences. In comparison to the work of Wolny et al., our simulation setup is arguably more accurate as united-atom approximation implemented in the CHARMM19 force field, which explicitly includes only the non-polar hydrogens, was removed and the fully atomistic CHARMM22 force field used. Nonetheless, we agree with the reviewer that the choice of the simulation setup should be better justified and have added this now in the Methods (p.20).

Also, we agree with the reviewer when s/he states that a simple estimate of the 813-813 distance could be obtained based on the available structure of the coiled-coil region and the known pitch distance of an α -helix. Indeed, by fitting perfectly straight, 70-residue long helices at the end of the coiled-coil region yields a predicted 813-813 distance of 25.2 nm, which is comparable (although not equivalent) to the average 25.7 nm value observed in MD. Nonetheless, our simulation analysis reveals an unexpected degree of flexibility in the context of the Myo10 dimer underlining both bending points of the SAH region when sandwiched by and IQ/CaM and hinges in the dimerization domain. We stress that this simulation was made possible by the crystal structure obtained in this work, which provides for the first time the boundary between the SAH and the dimerization region. Also, we note that the SAH region in Wolny et al had been analyzed by taking the helix of the dimerization region as part of the SAH. Although this may be correct if Myo10 acts as a monomeric activated motor, this is clearly incorrect in the context of dimerized Myo10. The MD relaxation of the crystal structure revealed not only that the fully extended conformation of the dimer is favored, which accounts for the very large steps the motor takes when actin binding sites are available for this, but also shows where binding would most likely occur, which is not at the junction between the proximal and distal SAH (with hydrophobic residues) or within the distal SAH, but more at the junction between the SAH and the dimerization region.

Frankly, we believe that both the relaxed structure and the dynamics of this critical hinge region between the two myosin heads could not be predicted without MD and have decided to keep the simulation results in the Main Text.

Note that prior to this study, no paper on myosin X had predicted it could perform large steps of up to 53 nm on bundles. Previous structural models of dimeric myosin X (from the Zhang lab) had proposed an atomic model that predicted that either 18nm or 36 nm steps would be performed, but did not account for the ability of myosin X to take large steps only when on bundles.

The authors then go on to discuss the single molecule motility of human myosin X. It is not clear to me why, despite the presence of an 'anti-parallel coiled coil', the authors still need to dimerize the shorter 'HMM' construct with a leucine zipper (albeit with a longer section of spacer sequence

between the anti-CC and the leucine zipper compared to previous studies). And they need to use a separate technique to dimerize full myosin X, by crowding the molecules onto actin filaments in rigor, and then releasing them. Particularly important here is that in previous papers where the authors have used this 'crowding' approach for myosin VI, data was provided on how many molecules were dimerized, and for how long. The same data should be provided here. It would also be highly informative to see images of the 'dimerized' molecules - do they look similar to the rare dimers seen by rotary shadowing EM published in ref 12 ?

The need to dimerize myosin X in this way reinforces the issue that myosin X only appears to dimerize fairly weakly. It is still not clear (despite quite a few papers on this subject) if it dimerizes in the cell, and if so, by how much? With only ~1000 molecules in the cell, can it ever become clustered enough to dimerize, or not?

Unfortunately, we did not correctly describe the technique used for single molecule studies with the full-length myosin X construct. This has now been corrected (**p.18**) with what should have been in the submitted manuscript. The methodology that was in the submitted version of the manuscript came from a much earlier draft and was in fact our myosin VI protocol. Early experiments had revealed that we had sufficient full-length dimers to conduct single molecule experiments without actin crowding, implying that the coiled coil of myosin X is of higher affinity than that of myosin VI, which is consistent with published EM results and published measurements of the Kd for myosin X dimerization. Thus we maintained full-length myosin X at high concentrations until diluted into the motility chamber. This provided an adequate number of moving dimers for our experiments. We have not performed EM experiments, but since the approach is much the same as the published EM work of Knight et al. (2005), we see no reason to do so. We anticipate that a similar population of dimers (10%) would be formed, which provides more than enough for single molecule observations.

The addition of the leucine zipper to our truncated construct was to evaluate whether or not the inclusion of a spacer between the zipper and the anti-parallel coiled-coil would allow the type of single molecule behavior we observed for the full-length construct on bundles, in contrast to the previous published work with zippered dimers. This was indeed the case for actin bundles, but not for single actin filaments. On bundles, we saw greater velocities of movement as well as a new population of large steps, as we measured for full length myosin X. We also wanted to have the leucine zipper for kinetic experiments to insure that all molecules were dimers, rather than a mix of monomers and dimers.

With such weak dimerization of the zippered HMM construct, can the authors be sure that the molecule is in fact dimerized through the antiparallel coiled coil, and not solely through the leucine zipper? If the helices that form the anti-parallel coiled coil dissociate, then molecules would only be dimerized through the leucine zipper, which in this case is placed further away from the SAH domain, compared to previous studies. Could this account for the longer 57 nm step in HMM compared to the full length construct (52 nm)?

As noted above, our observations as well as earlier published results suggest the affinity of the myosin X coiled coil to be in the μM range. Thus given that the effective concentration will be much higher than this in the zippered constructs, it is highly unlikely that the coiled coil is not formed in the presence of the zipper. The fact that we see gating implies that the two heads are not simply tethered by the leucine zipper with a long and flexible linker instead of the anti-parallel coiled coil. However, it is possible that a few residues after 938 help stabilize the anti-parallel coiled-coil. These residues are missing in the HMM, which may allow occasional, partial unzipping of the anti-parallel coiled-coil in the zippered dimers. This may be the basis of the slightly larger steps measured for the zippered dimer, as compared to the full-length molecule. However, this larger step size was not in fact statistically significant. These points have been added to the discussion (**p.13**).

Can the authors show the raw data for the histograms presented in Fig. 2? E.g. superimpose a number of frames from a time series and show the image, or some other approach?

We have added such a figure to the supplement (**Supplementary Figures 3-4**).

The data on movement on individual filaments and bundles is interesting (Fig. 2), but I'm not convinced that it makes a radical new development compared to previous data already published from the Rock group and others (Table 1). The step-size plots are selectively over-interpreted to fit the step sizes that generated by their model. All the backward steps are placed into a single 'bin' but not the forward steps. Separate peaks are 'seen' in the histograms for the bundled data, but a slightly different/adjusted analysis of the single filament data, could also show multiple step sizes from eyeballing the data. No statistical analysis is provided. No details are given of the errors (e.g. mean +/- S.D.) for these data. For these reasons I recommend that this data is moved to the supplemental material, and careful consideration is given to these criticisms.

The step size plots were not fit to a manually chosen number of populations of steps. We provided the best fit to the forward stepping data based on unbiased fitting, as we now show and discuss in **Methods p.19** and in new **Supplementary Fig.4**. Reference to the fitting algorithm was omitted from the submitted manuscript during shortening to fit within the journal word limit, but has been added to the supplementary materials of the revised manuscript. The methodology used can be found in: Benaglia et al., Mixtools: An R Package for analyzing Mixture Models. *J Stat Softw.* 32(6), 1-29 (2009c).

While the data is suggestive of multiple populations of backward steps, the backward steps were under-sampled and could not be reliably fit. This was the reason we fit them as one population. We now state this and add two population fits for the one data set where such a fit was somewhat better than a single exponential fit (HMM on bundles. The size of the backward steps is not of as much importance as the size of the much more numerous forward steps.

The overall importance of the contribution of the step size histograms (for the full length and zippered HMM constructs) is to illustrate that a population of large steps that were not shown in any of the earlier studies is present. We feel that the earlier dimerization strategies did not allow the anti-parallel coiled coil to form properly, which prevented the largest steps, as well as the high velocity movement on actin bundles. This is an important point of the paper and should remain in the main body of the paper. The step size data with optimal were kept in the body of the paper as it critical to support the model that comes from the single molecule data combined with the structural data. As requested by the reviewer, the run length figures have been moved to the supplemental material. The results are summarized in a table (**Table 1**).

Similarly, the assertion that the two-headed construct is 'gated' needs more data, rather than just the difference in ATPase rate. I'd like to see the results for MantADP etc (as performed by the Howard White Group). And what would happen to these measurements if bundled actin, rather than single filaments were used?

Doing kinetic experiments in the stopped flow using actin bundles has not been previously reported by Howard White or any other group. However, we have gotten this to work and have added data to the paper that demonstrates that the rate-limiting step for the myosin X ATPase cycle on actin is the transition that is measured by the change in pyrene-actin fluorescence. This rate is about 19/sec for single headed myosin X on either single filaments or on actin bundles. For the zippered HMM, this rate is also ~19/sec on single filaments, but increases to ~34/sec on bundles. This increase in the

rate-limiting step underlies the high velocity of the myosin X dimers on bundles. This has been added to the results (p.7-8 and Table 2) and discussion of the paper (p.12). A kinetic figure (Supplementary Fig.6) has been added to the supplementary data.

For me, and for the field, the most interesting part of the results comes in the next section on the structure of the myosin X motor domain, although this section could be written much more clearly. Nter is not defined (for example). The paragraph needs breaking up into two, with a second paragraph starting on 'the pre-powerstroke structure'. And all this data (currently in supplementary figure 4a,b) needs to be in the main paper, and this section really expanded so we can see in detail how the converter region, and the Nter interaction are important in this motor, and different to that of other myosins. I therefore strongly recommend that this data is given much more prominence in the paper, and rescued from its position mainly in supplementary, and that the stepping data is placed in supplementary data as it is less informative and less novel.

The main text was changed as proposed by this reviewer (p.8). We have now defined Nter in the text. We moved the previous Figure 3 on run lengths to the Supplement. As suggested by the reviewer, we have now included another main figure : Fig.3 to present the pre-powerstroke of Myosin X and in particular the position of the converter and the interactions with the motor domain that allow the stabilization of its position. As well as the difference of the SH3 between myosin Vc and Myosin X.

Hidden also in this one paragraph/section is 'we also crystallized motor domain of Myosin Vc in this state' to compare the structure of Myosin X with that of Myosin V. But this is surely not the case, as there are multiple structures of Myosin Va already published! It seems to me that Myosin Vc structure has been solved by these authors, and it has been inserted into this paper as a way of 'publishing' the structure more easily! Moreover, there are no details on the construct, in the methods, how it was made etc etc, and the sudden arrival of a structure of Myosin Vc is surprising. I suggest either removing this structure, or more strongly defending its inclusion. This becomes even more confusing, when it's myosin Va that is compared to MyoX in figure 4c!

While there are several myosin Va structure available, none of them are in the pre-powerstroke state. To compare how the PPS of myosin X differs from that of myosin V, a structure of the PPS of myosin Vc and Vb was attempted. We were successful for the Myo5c motor domain, which is reported here. Note that a recent publication (Wulf et al PNAS 2016) has reported a crystal form of the MyoVc PPS that closely resemble this structure but differs due to the fact that the two molecules in the asymmetric unit differ in the exact position of the converter. We thus decided to include this data here since it allows a fuller comparison with the myosin X pre-powerstroke. We now introduce this MyoVc structure more clearly in the main text page 9. As stated above, we could not get a Myosin Va PPS structure. Thus we are left with comparing myosin X in PPS with MyoVc, while the data recently obtained for myosin Va in rigor (Wulf et al PNAS 2016) allows us a direct comparison with the rigor state of Myosin Va. Unfortunately, one cannot always choose which isoform or which species one can use successfully for crystallization. This is now presented p.9.

Next, the authors investigate the rigor state of myosin X on actin (Fig. 4). The legend to Fig 4 states that they see the first 3 turns of lever arm helix, but this is not labeled on the figure - is this a spike of density sticking out of the converter? Super-imposing an atomic ribbon model on the EM density, makes it impossible to see the EM density. The authors should show a panel first without the model. In addition, supplemental figures 5b-f should be moved to the main part of the paper.

We have now changed Figure 4 (New Fig.3 and Supplementary Fig.8) and its legend to make clear that the construct only contained the converter in which the last helix was directly visualized and to

present the EM data more thoroughly as requested by the reviewer. In particular the density of the converter is shown distinctly. A movie is also included to assess better the quality of the CryoEM data.

The authors also comment that 'the SH3 domain is different' to other myosins, but do not go on and further explain how it is different from that seen in previous structures. They should.

We have now incorporated a panel in Figure 3 to show how the SH3 (**Fig.3c**) domain differs and also in Figure 4 to show how the SH3 domain as found in myosin V would generate clashes and is thus not compatible with the lever arm position found for the rigor state of myosin X.

Finally, the authors need to clarify what the C-terminal residue is of the construct used here in cryo-EM, so that we can tell how long a section of lever (which ought to be the first IQ motif) we should be expecting to see.

We have now made this clear in the text and the figure legend (**New Fig.3 and Supplementary Fig.8**).

In the last section of the results, the authors go on to model the dimer (Fig. 5a). However, the diagrams of actin shown (Fig 5c and 5b) are so wrong that it undermines my confidence that their conclusions are correct. Actin is shown as having two chains of subunits that are axially aligned rather than offset by a half subunit, and which are left-handed helices rather than right handed. These are really basic errors that are out of place in a paper on acto-myosin structure. They must replace these with correct ones, and revise their conclusions if necessary. The justification for the axial and lateral offsets they have used between adjacent actin filaments in a fascin bundle, which allows them to calculate distances between hot spots is not referenced in either Fig legend, but it should be. I'm also confused by the say that the myosin molecule is drawn, which seems to be more consistent with parallel dimerization, than anti-parallel (e.g. compare fig. 5b - which is clearly antiparallel, with fig. 5c, which is not).

We are embarrassed by the mistake that was introduced in the cartoons describing the actin helix and have now corrected **Fig.5c** and **Supplementary Fig.11**. We agree with the reviewer that these should have been drawn to reflect the correct handedness of the actin helix and this is now done. However these cartoons were not used to derive our conclusions on how these myosins walk. This was done from careful analysis with structural models of F-actin and myosin heads as shown in Figure 5d. The cartoons were done too quickly to just summarized what had been the conclusions from these careful analysis on what distance between the 3IQ region of a rear and lead heads bound to different actin subunit would be compatible with the distance found possible to be explored by the SAH-cc-SAH.

We have now added also in the figure legend the reference (**ref 45**) that we used to calculate the axial and lateral offsets for the fascin bundle. Using AFM and EM, these authors (**ref 45**) report 9 nm between actin filaments and their hot spots seen by AFM (Fig.1) indicates the lateral offset as well as the distance between hot spots that have been estimated as 19 nm and 52 nm.

The representation of the dimerization of myosin X tries to mimic the dimerization domain, which consists of an interaction of two short helices for each heavy chain. See for example Fig.1c or Supplementary Fig.11c.

Moreover, in Fig 5d, the X's and vertical dotted lines and arrows, should have equal increments of distance of 5.5 nm (as implied by the integer values in nm put next to them), but they don't. Instead they get closer together at the right hand side, so that the 57nm one isn't pointing to the last motor at all, it's pointing at the preceding one. This is out of place in a paper on structural biology.

In fact, for the previous figure, we had chosen a characteristic element of the motor domain: namely the base of the HCM loop of the motor to draw the arrows and indicate the distance between myosin heads. We can see how this can confuse the readers and we have now changed this (Fig.5).

Discussion.

The first paragraph returns to the 'dimerization' discussion, and goes back to stating that the 'dimerization region of myosin VI precedes the SAH', but again, the authors should cover the controversy in this field regarding this statement. And this needs to be addressed at later stages in the discussion.

We have added sentences to describe this controversy and cited corresponding papers (p.11).

The assertion in the discussion that 'the 34 residues after the consensus SAH' needs to be rewritten, as these residues were already shown to form a SAH (refs 12 and 13) and this new data only confirms these earlier observations. Moreover, as explained earlier, there is no 'consensus SAH' and known SAH domains can contain hydrophobic residues.

We have rewritten this part in discussion p.11.

The short discussion on selectivity for actin bundles, does at least concede that this selectivity has been shown before. But I'm still not convinced by how novel the data presented here is. And it's not clear if the anti-parallel coiled coil can still form in HMM when the molecule is dimerized by a leucine zipper as no data is shown to support/discount this idea. And no images are shown of the 'clustered' dimers of full-length myosin. And lastly, it is completely unclear to me how leucine zippered constructs reported in earlier publications could 'somehow shorten the lever arm'. What would the mechanism be?

The mechanism of shortening the lever arm would be by having the distal SAH region participating in dimerization of myosin X. This explains the data reported by Rock et al in 2008 and his recent but unpublished structural studies demonstrate that indeed this proximal region was engaged in dimerization in the context of the artificial construct that he had studied in this publication. This explains the discrepancy in the single molecule experiments. It also highlights the value of the current study, which has the merit of reporting for the first time how dimers of full length Myosin X without any bias do walk selectively on actin bundles. Note that the previous construct from Ron Rock reported selectivity for a Myo10 chimera, not the native Myo10. This finding of selectivity was essentially an accident because the artificial dimerization of their chimera promoted steps of 18 nm, which cannot be performed on single F-actin filaments, as we noted in the Discussion. Note also that studies from the Goldman/Ikebe and Sellers/Molloy groups studying different artificial Myo10 dimers did not conclude that Myo10 was selective. Their chimera did not conserve the proper dimerization of native myo10 and this is apparent also in the fact that they did not observe the larger step sizes, nor the accelerated velocity on bundles, that we are now report to be possible for both the Myo10 full length and the Myo10 HMM.

We again apologize that part of the methods were incorrectly described which prevented the reviewer to fully understand the experiment. We did not cluster the full-length molecules on actin, which could promote aggregation. We reiterate that we do think that reporting the stepping behavior of full-length myosin X molecules is an important contribution in this paper. In particular since the structures help explain how different stepping can be achieved on single actin filament and on bundles. Note that a population of large steps (52-57nm) have never been reported for any myosin to date and they are directly related to the special structural features of the myosin X motor.

I recommend that the discussion places more emphasis on the structure of the motor. The discussion would benefit from shortening, and more clarity, and discussion of the following: The stiffness of the SAH domain is probably about 10fold lower than that of the IQ lever (e.g. seem data from Spudich group in PNAS), and one might expect the SAH domain to be more flexible - how would this affect the data interpretation? The majority of myosin X in a filopodium is actually found at the tips. Do the authors think that the re-modelled converter/lever that they have found in myosin X might be important for linking myosin X to the ends of actin filaments at the filopodial tips?

Concerning the flexibility of the SAH, what our modeling suggests is that the largest steps performed by myosin X on bundles must come from an overall conformation in which the IQ-SAH helices and dimerization region are all straight, allowing the heads to reach the maximum length possible for this myosin on actin bundles. As we have discussed in the text, we believe that the other steps correspond to pliancy (flexibility) between the dimerization domain and the SAH rather than flexibility within the SAH, which would likely results in a larger proportion of (18-25) nm short steps than what has been observed.

While we discuss how this geometry of myosin X could be well adapted to trafficking in the filopodia, very little is known about how myosin X interacts with actin filaments at the tip. However we can speculate that geometry would clearly help a single, dimeric myosin X crosslink filaments at the end of filopodia and regulate their dynamic growth.

Methods

Is there a reason why the human constructs are used in some experiments, but the bovine Myosin X was used elsewhere?

There is no particular reason except that we have worked with constructs that allowed us to get crystals. It happened that the first crystals obtained for the motor domain were from a human construct produced from baculovirus/ SF9 cells but that we had previously obtained crystals from a bacterial-expressed protein for which the DNA had been ordered based on the sequence of the bovine myosin X sequence for the SAH-cc region.

Dimerization of myosin 10 by clustering - needs more detail (see above). The authors observed dimerization without actin clustering 'althouth (sic) in a smaller fraction', but do not give any details (again see comments above). Were these non-clustered dimers ever used in experiments?

Again, we apologize. Clustering was not used, as there were sufficient numbers of dimers formed without clustering.

Crystallization: no details are given for the MyoVc MD construct until this section. They should be (or MyoVc data removed)

This was added earlier in methods.

Molecular dynamics

No details given on how the force field was chosen, they should be

We are sorry for this omission – we have now added this in methods.

Cryo-EM Methods section

No mention of having any myosin in the frozen sample, nor what concentration of actin they used, nor anything about the composition of the solution that was frozen is provided. The solution termed 'EM' buffer has no buffering species in it, which is hopefully wrong as the pH could be just anything! What construct of myosin- X was used for the EM, what buffer was it in prior to use for EM and what was the composition of the mixture once the S1 had been mixed with the actin before freezing? The text in Results and the Fig 4 legend suggests that a motor domain alone (i.e. no IQ motifs) was used, but there is no mention of such a construct in the Methods section. Did they use the whole S1, but the lever was disordered in the EM or what?

In the submitted manuscript, we accidentally omitted the buffering species itself from our description of the EM buffer, and we have now corrected the description to include the 5 mM MOPS (pH 7) that was used. We have also elaborated on the description of the sample preparation for vitrification – for example, we now describe the process by which actin was mixed with S1.

A description of the truncated myosin X construct (residues 1-754, motor domain only) is also now included in the first section of the Methods section (p.21).

Table S2 mentions crystal structures of MD alone (and other things too (including some myosin-5 constructs), but there is no mention of making these in Methods, so what are they - what residues do they encompass?

In fact it was reported as Myosin MD Vc – but we have now rectified it for Myosin Vc MD

A proper Methods section that fully describes (and names) every construct used in the whole study should be provided. These names should be used consistently in the rest of the paper so that there is no ambiguity over which data come from which construct. A Table would help. While Fig S1b is helpful, it does not cover everything that appears to have been used. It would be impossible for anyone to reproduce their work from the information they have given.

We have corrected what was not correctly described in the earlier version.

Overall, in both the main paper, and in supplemental, there are a lot of typos, etc, so it needs a proper edit. E.g. Fig S3 legend says they've used a monomer truncated prior to the coiled coil. Which bit of coiled coil, so what is the C-terminal residue? They should have a name for the construct in Methods, and refer to it by this name in this Fig legend. Further on in the legend they refer to wild type and HMM as having a single Vmax. Where is the data for wild type or this this a typo? Then later "S1 was and 19.6" - means what??

We are sorry that indeed the supplemental and the methods contained lots of typos and missing information due to an error that has been introduced in the version of the file that we shared between us. We are sorry not to have checked carefully that all corrections previously introduced were all in the final version prior to submitting it.

Other comments:

1st para of results mentions 'gating' but does not explain what this is, or provide any references.

We added a sentence and reference to explain gating to this paragraph (p.5).

The SAXS data on the CC-PH1-PH2 seems really redundant to me. While it may show that this construct is extended, I don't see that this data, on its own, supports the idea that the antiparallel CC is not influenced by the C-terminal sequence as claimed. I recommend that this data is removed. If not removed, the SAXS (Supplementary Fig. 2a-d) and modelling data (Supplementary Fig. 2e-g -) should be presented in separate figures.

We have removed this data since it is likely less important than other messages in this paper and to shorten the manuscript.

A recent paper on myosin X in prostate cancer was published in cell reports (doi: 10.1016/j.celrep.2015.11.012) has been omitted from the introduction (1st paragraph).

This was added to the manuscript (**ref.11**)

Reviewer #2:

Ropars et al. present a concise study of myosin X. Combining X-ray crystallography and cryoEM, the authors determined structures of the pre-powerstroke and rigor state of the myosin X motor domain and describe an extreme angle of 120° for the powerstroke. They determined a crystal structure of the dimerization domain including the SAH domain. SAXS data of the full-length protein show that it is flexible in solution. Single molecule studies with new constructs including the dimerization domain allow a more accurate measurement than those that have been performed previously by others and show that myosin X has a different stepping behavior on single F-actin filaments and filament bundles.

The authors excellently discuss their results. Their findings are important and should be published.

We thank reviewer 2 for his interest in our story and for his very careful reading and corrections for the manuscript.

However, the manuscript has the following shortcomings:

1. *In their introduction and in the rest of the manuscript they do not clearly describe what was there before and has been published previously by others. For example, Lu et al. presented a NMR structure of the anti-parallel coiled-coil dimer region of myosin X in 2012. It is not mentioned in the introduction and in the Results part, Ropars et al. present it as if it has been unknown so far. The only new aspect of the crystal structure is that the linker is a helix as proposed previously (see review by Li et al. 2016).*

The introduction to previous work has been done clearly now in the introduction and within the manuscript (p.3 ; p.5 ; p.11). We are sorry that the first version didn't cite earlier the anti-parallel coiled-coil contribution from Lu et al. However although this paper provided a key element, it was not clear it was not an artifact at the time since coiled-coil out of their context can form very different structures. It was thus still critical to get information of the dimerization region in the native context to assess for the structure and dynamic of the lever arm/dimerization region which provides a key element for how the two heads can step and why they prefer very large steps on bundles.

In the review Li et al 2016, the authors state that the straddling mechanism for myosin-X has not been demonstrated so they agree with the fact that further work was required to demonstrate the selectivity of native myosin X walking. In Figure 5, the authors propose a model without further data compared to what was published in 2012. They cannot predict correctly that the structures we have now from work in this study concur to predict the fact that myosin-X could take 52-57 nm steps, rather than just 36 nm steps as previously measured with chimeras.

2. The only result from the SAXS data is that myosin X is flexible as full-length protein. This is not surprising. The models based on the SAXS data as presented in Supplementary Fig. 2 are pure fantasy.

We agree with this statement. The experiment was meant to get data against the hypothesis that the region after dimerization would provide more define structural organization of the tails. This is not an essential element and we removed this data, as proposed by the reviewer.

3. The pre-powerstroke structure of the motor domain of myosin X is very similar to that of other myosin motor domains (Vc, VI, Ic, smooth muscle myosin). It is nice to have it, but unfortunately it does not reveal new important details.

In fact, while the motor domain is similar, the important details are in the position of the converter, which differs from that of the other motors as described in the text. This information was not

sufficiently highlighted to be noticed – it was in fact presented in Supplementary Fig.1 while figure 4 assembled too many important structural results. As proposed by reviewer 1, we are now adding clearer figures (**Fig.3 and 4**) to highlight what the new structures provide. In particular, we present clearly the structural adaptations that explain why this myosin can take such large steps.

4. By actual measures, the cryoEM structure is of low resolution although the authors used a direct electron detector for recording their data. This is surprising and makes the proper fitting of the converter domain - a central aspect of the manuscript - questionable.

It turns out that the FSC computation we have performed for the original submission underestimated the resolution of our cryo-EM structure. This was due to a subtlety in the way our reconstruction was performed. We did not apply helical symmetry to the final volumes, which means that the resolution is worse at the ends of the filament (i.e. near the edge of the volume) due to image misalignments being amplified in these regions. However, the mask we initially used for FSC computation included the entire filament, so that the FSC resolution estimate was 'contaminated' by the peripheral ends of the filament where the resolution degraded.

When we re-computed the FSC for only the central four actomyosin subunits in the map, using a soft-edged mask, the resolution estimate increases to 9Å. We emphasize that we are using a highly stringent "gold standard" for this FSC computation– in which the two volumes being compared were reconstructed completely independently from start to finish, using different images. This stringency, along with careful application of soft masks during the FSC computation, causes our FSC estimate (0.143 criterion) to closely mirror the level of detail visible in our maps, as has been seen in numerous published cryo-EM studies. Unfortunately, the map quality was not well exhibited in the figures presented in the earlier version of this paper. We have therefore generated new figures (**Fig.3 and Supplementary Fig.8**) and **movies** that properly highlight the visible detail in our maps. See also below.

Although the resolution achieved for the system remains low by modern standards for cryo-EM, it was certainly sufficient for the purpose of defining the orientation of the converter domain (see below). In fact, our samples suffered from numerous problems that doubtless hampered our ability to increase the resolution further, including (1) poor ice quality, (2) poor binding of the myosin construct on actin, leading to relatively low occupancy of the myosin sites; (3) high required concentration of actin/myosin required to achieve even moderately good binding, which lead to increased contamination of the imaged signals, due to overlaps with neighboring actin and myosin molecules; additionally (4) we observed evidence of substantial heterogeneity in the filaments themselves, a known property of actin.

Despite the above issues with our sample, we note that the resolution achieved here is comparable to other recently published actomyosin structures. For example, the recent myosin V actin rigor complex cryo-EM reconstruction (Wulf et al, PNAS 2016) shows a comparable level of detail as our structure – despite the fact that Wulf et al. used a better microscope (Titan Krios with an energy filter, vs. our F20) and collected two orders of magnitude more data (3,725 images vs. 53 images for our current paper; 380,000 asymmetric subunits vs. 20,000 in the current work); this difference can be attributed to our use of a K2 direct electron detector, vs. a CCD detector in the Wulf et al work.

5. When presenting their model, the authors do not state that others have proposed the model before (see for example Lu et al. 2012).

While the earlier model of Lu et al. (2012) did incorporate an anti-parallel coiled coil, as does our model, the major difference is that the model of Lu et al incorporates semi-rigid connections instead

of the highly flexible connections and greater lever arm swing of our model. The previous model can account for steps of ~ 18 nm and ~ 43 nm, but not the larger steps that we observe. This has been added to the discussion (p.14).

Taken together, Ropars et al. confirm and extend known data and present a good explanation for a model that has been previously described by others, for example by Lu et al. 2012. Nevertheless, their findings are important and should be published after the named issues have been resolved.

We thank this reviewer for his positive assessment of the work. We have clarified what was known with the previous contributions and how the current paper now establishes how myosin X walks with step sizes larger than ever measured for any myosin. This had not been really anticipated before although homology models of myosin X had been proposed. Furthermore, the paper now incorporates new kinetic data that explains how myosin X dimers move with higher velocity on actin bundles as compared to filaments (p.7-8).

Other specific points:

1. The authors give exact values without deviations for all measured step sizes although the measurements do not allow for this accuracy. Proper deviations should be given.

This has been added.

2. "...reconstruction was ~ 10 Å..... sufficient to unambiguously define the position and orientation of most alpha-helical elements in the map." A map at around 10 Å is definitely not high enough to unambiguously define the position and orientation of alpha-helices.

As discussed above, the resolution in our maps is now confidently assessed at 9Å after we corrected our FSC calculation, and this new estimate is supported by our ability to resolve alpha helices. The **new Fig.3** as well as **Supplementary Fig.8 and Videos 1 and 2** provide evidence for these features.

3. What is the resolution of the converter domain in the cryoEM map? Is the resolution high enough to unambiguously fit the model in this region. I suspect that a detailed fit of the last helix of the converter domain and determination of the degree of the lever arm (Fig. 4) is most likely not possible.

These are excellent points. The resolution of the converter domain is certainly less than of the rest of the motor domain. We have performed a masked (gold-standard) FSC calculation and find the resolution of the converter domain could be as low as 11Å (shown in the **new Supplementary Fig.8**). We note however that the gold-standard FSC probably under-estimates the resolution of our final map (generated from all the data), because densities for the three alpha helices in the converter domain appear to be well resolved. It is possible that the alignments in our 'gold-standard' refinements didn't converge as well as the full-data-set refinement, due to the smaller number of particles; we suspect that this effect causes our gold-standard FSC measurement of the converter domain to underestimate the resolution.

Given the uncertainty in the resolution in the converter domain, we decided to further investigate whether rigid-body fitting gives a consistent solution in this case. We therefore switched to an exhaustive search method (from the Situs package) and independently performed fitting calculations on both the half-data-set reconstructions as well as the full-data-set reconstruction. **These calculations are fully consistent with each other, and with our earlier fitting results from UCSF chimera.** The highest-scoring solution in each case clusters around the novel position/orientation we have identified that is rotated $\sim 30^\circ$ with respect to the rigor state of myosin V and other myosins.

These new fitting calculations represent a strong test of the hypothesized converter domain position, since the half-data sets were refined fully independently of one another, each starting from a highly-filtered (40Å resolution) reference model. These results are presented in the new **Supplementary Fig.8**.

As the reviewer notes, it would be difficult or impossible to uniquely define the orientation of the lever arm helix by itself in our maps, at the current resolution. However, the additional secondary structure elements of the converter (two alpha helices and a beta sheet) provide sufficient features/mass that a unique solution can be found. This is true even for our half-data-set maps, where the alpha helices in the converter are no longer clearly resolved— as evidenced by the consistent results found for both of these maps (please refer to the new **Supplementary Fig.8**) Based on a comparison of these fits with each other, the orientation of the converter (and thus the lever arm helix) is defined to an angular precision of +/- 5° or so. **Note also that the new position of the converter is accompanied with a large shift of the relay density and this is also very consistent with this new orientation for the rigor position of the lever arm in this myosin.** We are therefore confident of our conclusions regarding the novel converter orientation identified here, and we thank this reviewer for pointing out the weaknesses in our earlier analysis.

Supplementary Figures 5b-e including legends are difficult to understand. Maybe the subfigures can be split in two, one subfigure showing the map/simulated map/envelope and one figure of the same field of view of the models. Which PDB was used for the myosin II model? Why 'Myosin II model' and only 'Myosin X' without 'model'? A movie could be used to make their statements clearer.

In order to address these deficiencies , we have completely redone Supplementary Fig.5, moving the important results into **Figure 3** in the main text, and added **two supplementary movies** as well.

4. For me it is not really obvious, how the authors used the 'Fit in map' function of Chimera to perform local refinements for creating the whole rigor model of myosin? It should be stated how the pre-powerstroke state model was fitted (rigid-body fitted into the cryoEM map is not sufficient, as the myosin structure differ in 1-670 between rigor and pre-powerstroke state and thereby is not rigid at all). Why have the authors not flexibly fitted their pre-powerstroke state model into their rigor density?

These are excellent points. We suspect that it is difficult or impossible to generate a meaningful all-atom model of the myosin X rigor state from our cryo-EM map at the current resolution, using currently available methods. Certainly the conformational changes involved in switching from the pre-powerstroke state crystal structure to the rigor state are very large and exceedingly unlikely to converge in a flexible fitting calculation. The main conclusion we draw from our map, at the current resolution, is that the converter domain is substantially re-oriented with respect to other rigor myosin structures. We are therefore sticking with rigid-body fitting methods, using appropriate X-ray models that illustrate this point. Thus, in our new **Fig.3**, we present a ribbon diagram of rigor myosin V fitted into our myosin X map. We don't claim this is a perfect model of the myosin X motor domain by any means; the purpose of the figure is to illustrate the very significant change in the position/orientation of the converter domain.

5. The authors describe a simple and plausible model for different myosin X step sizes. Is there any evidence how fascin binds and builds up the F-actin bundles? The authors assume a flat and organized structure. Is it not more likely that the F-actin-fascin bundles are less flat or less well organized in vivo? The broad distribution of possible step sizes is likely an indicator for that. This aspect should be discussed.

As state in the text and shown in Supplementary Fig.9, the model for the fascin bundle we have used to interpret our data comes from a prior AFM study (ref. 45 : *Ishikawa, R., Sakamoto, T., Ando, T., Higashi-Fujime, S. & Kohama, K. Polarized actin bundles formed by human fascin-1: their sliding and disassembly on myosin II and myosin V in vitro. J. Neurochem. 87, 676-85 (2003).*)

6. It does not make any sense to give values like: 32.72 +/- 14.27. A decimal number is not meaningful at all in this case and 33 +/- 14 would be correct.

We agree, and have corrected this (p.6).

7. Measurement deviations of SAXS data are completely missing.

We have now removed the SAXS data as proposed.

8. Pymol is not cited.

This reference has now been added.

9. The statement that their work 'may also reveal targets for therapeutic interventions to combat metastatic cancers' is a bit overstated and should be removed.

We removed this.

10. Methods are very rudimentary and one would not be able to reproduce their results based on the descriptions.

We have added details and references that completely describe the methods used.

11. Fig. 1b/c rotation between the two views of the same models should be indicated by an arrow and angle.

Fig.1b/c has now been corrected accordingly.

12. p. 5 "...rather than partially structured linker as previously described". This statement is not correct. Lu et al. describe this linker as "semi-rigid helical linker". This is what the authors in principal show with their MD simulations (Fig. 1e).

We removed this.

13. What is ~nanometer resolution?

As described above, we have improved and expanded upon our original resolution estimates, and present separate FSC curves for the actomyosin complex, the actin region alone, and the converter region alone - yielding estimates of 9.0Å, 8.8Å and 10.8Å respectively. These are detailed in the new **Supplementary Figure 8**.

Reviewer #1 (Remarks to the Author)

This revised paper nicely shows the SAH domain/anti-parallel coiled coil in myosin X and its role in bundle specificity, together with a novel structure for myosin X motor domain that demonstrates the large movement of the lever between pre- and post power stroke states, together with the novel position of the lever/converter compared to myosin V. It makes a nice contribution to the literature, and in this revised paper, there has been a significant improvement in the presentation of the data and discussion of the results. There are still a few typos/grammatical errors but I'm sure these can easily be remedied.

a very few minor comments:

On line 63, page 3 in the introduction:

It should be made clearer that reference 13 measured the length of the SAH domain from EM images for both monomers and the rarer dimers;

"For myosin 10, the mean head lengths were 34.7 +/- 4.4 (S.D.) nm (n = 27) and 32.7 +/- 3.9 nm (n = 28) for monomers and dimers, respectively. These values are much longer than the expected head length of 18.4 nm, based on 8 nm for the motor domain and 10.4 nm for the three calmodulins",

Therefore, I think this sentence should be rephrased to add, 'and by measurements of monomers and dimers of Myosin X HMM constructs from rotary shadowed EM images.

For the SAH domain measurements in this paper:

If residues 814 - 880 form a SAH, then at a rise per residue of 0.15 nm, the length of the SAH should be 10.05 nm, whereas what is measured here is 10.5nm - a discrepancy of 3 residues, can they confirm the length, and/or explain this discrepancy?

For discussion: The majority of myosin X in filopodia accumulates at the tips of the filopodia. Do they envisage that these molecules are only attaching to actin filaments when at the tips with just one head? (the tails are presumably attached to integrins). It would be interesting to see their thoughts on what is happening here, after all myosin X only moves along the actin bundles in the filopodia to get to the tips.

Labelling on supplementary figure 2 - too small to read easily

Reviewer #2 (Remarks to the Author)

The authors addressed most of my concerns and the manuscript improved accordingly. In particular, the overall resolution and quality of the fits are now well presented in the figures and movies. I still have some issues that should be addressed prior to publication.

I am wondering why the authors do not mention the EM density map under "Accession numbers". Are they planning not to deposit the density?

With their new Fig. 3 and 4, the authors tried to show and clarify the differences between the pre-powerstroke states of different myosin classes. I suggest that some assumptions should be relativized and/or clarified:

1. What is red and yellow in Fig. 3d?
2. Which MyoVc lever arm/converter position is given in Fig. 3b in comparison to Fig. 3e.
3. It is important to state that the overall position of the lever arm differs within the unit cell of a crystal (Fig. 3e) in the range of half the angle to the given extreme lever arm position of the myosin X in the pre-powerstroke state position. Because of this flexibility, the overall accuracy of angle determination should be treated with caution. Is it possible that myosin X crystallizes with a

lever arm position close to myosin Vc (light magenta)? The authors state on page 8 that they obtained the structure with different crystal packing, but they should at least mention the possibility that the packing could influence it. Furthermore, the authors describe this difference as "slight" but in comparison to the myosin X pre-powerstroke position it is almost half the way!

4. Fig. 3f: An arrow pointing to the barbed end is misleading. A double-headed arrow would be better or it should be mentioned that the arrow is illustrating the direction of the myosin movement.

5. Fig. 3g: Which rigor-like MyoV pdb was used? Ref 42?

6. Fig. 3h: As mentioned above, the accuracy of the angle should be treated with caution, as the pre-powerstroke state was shown to be quite flexible.

7. Fig. 3i: There is no red.

8. Fig. 3j: Low-pass filtering to indicate an overall position is used to emphasize a different position of the SH3 domain, but by performing this, it appears that the SH3 domain of the model does not fit the density. Furthermore, I would expect some density for the rest of the lever arm. Is it visible at another threshold without masking?

9. Fig. 4: Which MyoV and how were the myosin motor domains superimposed?

This sentence is still there: "...reconstruction was $\sim 10 \text{ \AA}$ sufficient to unambiguously define the position and orientation of most alpha-helical elements in the map." I do not question that the model is correct or the fits are right. However, a map at around 10 \AA is definitely not high enough to unambiguously define the position and orientation of alpha-helices. I suggest to write "we could fit" rather than "unambiguously define the position and orientation".

The new method section describes the applied experiments and computational analyses in greater detail, which improved the quality of the manuscript significantly. Some points (especially on the model building of the rigor state of myosin X) need still to be clarified:

How was the myosin X MD in the rigor state created (page 22)? Homology modeling? It is only stated that 1OE9 was used. Afterwards, only "Fit in Map" (rigid-body) was used for the motor domain? If it is possible to "unambiguously define the position and orientation", then I would expect that a flexible fitting approach was used?

Same paragraph: Which atomic model of F-actin was used?

The fitting procedure is still not clear, too. The authors state in the main text (page 10) that the converter domain was fitted as a rigid-body but in the method section they describe a more exhaustive way with the Situs package. Furthermore, they should add, when they only used a rigid-body fit (I guess, when they applied 'Fit in Map' in Chimera?).

I do not understand, why two randomly chosen halves should result into two different conformations of the converter/lever arm position.

Please check the references in the Figure legends. I am a bit confused, since the authors state in one of the legends: "The parameters for the actin bundle were derived from ref 50". Is that correct?

It is obvious from Fig. 2b and d that the steps do not have uniquely defined sizes and more a likelihood distribution. This should be addressed as mentioned before because the bundles are not perfectly flat and ordered.

Reviewer #1:

Reviewer #1 (Remarks to the Author):

This revised paper nicely shows the SAH domain/anti-parallel coiled coil in myosin X and its role in bundle specificity, together with a novel structure for myosin X motor domain that demonstrates the large movement of the lever between pre-and post power stroke states, together with the novel position of the lever/converter compared to myosin V. It makes a nice contribution to the literature, and in this revised paper, there has been a significant improvement in the presentation of the data and discussion of the results. There are still a few typos/grammatical errors but I'm sure these can easily be remedied.

We thank the reviewer for his interest in our story.

a very few minor comments:

On line 63, page 3 in the introduction:

It should be made clearer that reference 13 measured the length of the SAH domain from EM images for both monomers and the rarer dimers;

"For myosin 10, the mean head lengths were 34.7 +/- 4.4 (S.D.) nm (n = 27) and 32.7 +/- 3.9 nm (n = 28) for monomers and dimers, respectively. These values are much longer than the expected head length of 18.4 nm, based on 8 nm for the motor domain and 10.4 nm for the three calmodulins",

Therefore, I think this sentence should be rephrased to add, *'and by measurements of monomers and dimers of Myosin X HMM constructs from rotary shadowed EM images.'*

We have changed the text as requested by the reviewer.

For the SAH domain measurements in this paper:

If residues 814 - 880 form a SAH, then at a rise per residue of 0.15 nm, the length of the SAH should be 10.05 nm, whereas what is measured here is 10.5nm - a discrepancy of 3 residues, can they confirm the length, and/or explain this discrepancy?

This statement is totally correct and the structure indicates as predicted the correct length per residue. However, the 10.5 nm that was mentioned for the length of the SAH in the text corresponds to the distance between residues 814 and 884, not 814-880 as previously mentioned. This explains the discrepancy in numbers. We have corrected the text and the figures accordingly.

E814 and E884 are indeed the correct boundaries between the IQ region and the dimerisation region, respectively. Note that E884 was also the first residue of the dimerisation domain in the structure previously published 2LW9. We thank the reviewers for his very careful reading of the paper that allowed us to correct the text and figure accordingly to define correctly the boundaries of the SAH in the Myosin X dimer.

For discussion: The majority of myosin X in filopodia accumulates at the tips of the filopodia. Do they envisage that these molecules are only attaching to actin filaments when at the tips with just one head? (the tails are presumably attached to integrins). It would be interesting to see their thoughts

on what is happening here, after all myosin X only moves along the actin bundles in the filopodia to get to the tips.

We added three sentences to the discussion to provide our opinions on this point.

Labelling on supplementary figure 2 - too small to read easily

We have changed the legend of supplementary figure 2.

Reviewer #2 (Remarks to the Author):

The authors addressed most of my concerns and the manuscript improved accordingly. In particular, the overall resolution and quality of the fits are now well presented in the figures and movies. I still have some issues that should be addressed prior to publication.

I am wondering why the authors do not mention the EM density map under "Accession numbers". Are they planning not to deposit the density?

We have now added the accession numbers for this deposition.

With their new Fig. 3 and 4, the authors tried to show and clarify the differences between the pre-powerstroke states of different myosin classes. I suggest that some assumptions should be relativized and/or clarified:

1. What is red and yellow in Fig. 3d?

In red and yellow is shown the connectors of the converter called Relay and SH1 helix – this is now added to the figure legend.

2. Which MyoVc lever arm/converter position is given in Fig. 3b in comparison to Fig. 3e.

The converter position of MyoVc chosen in b correspond to the closest one to Myosin X. We added this information in the figure legend.

3. It is important to state that the overall position of the lever arm differs within the unit cell of a crystal (Fig. 3e) in the range of half the angle to the given extreme lever arm position of the myosin X in the pre-powerstroke state position. Because of this flexibility, the overall accuracy of angle determination should be treated with caution. Is it possible that myosin X crystallizes with a lever arm position close to myosin Vc (light magenta)? The authors state on page 8 that they obtained the structure with different crystal packing, but they should at least mention the possibility that the packing could influence it. Furthermore, the authors describe this difference as "slight" but in comparison to the myosin X pre-powerstroke position it is almost half the way!

The difference in angle for Myosin Vc is not large compared to the difference with Myosin X : we measured 14 degrees between the two myosin Vc positions, and there is a larger displacement of the converter and a different angle for myosin X that leads to ~50 degrees. The fact that the figure shown in the paper is not in 3D is a disadvantage to allow this visualization but the coordinates in the PDB will allow scientists to appreciate this.

We agree however that the accuracy for the position of the lever arm is not strict and we explain so in the manuscript also by introducing that the pliant region add additional flexibility for its position.

The different crystal forms of Myosin X in PPS show that there is no influence of the packing on the position of the converter which is controlled by the specific interactions we describe with the Nter subdomain of the motor domain in Myosin X and which are found in two different crystals with two molecules per asymmetric unit each. In contrast, Myosin Vc doesn't make such specific contacts between the motor domain and the converter and thus has a converter more flexible that can adopt different position. Note that if myosin X would not make these specific interactions and would adopt position of its lever arm closer to those found for myosin Vc, then the step distribution should be smaller. There is no influence from the packing in the case of Myosin X. The flexibility of myosin V allows two positions to be privileged but Myosin V has no interactions with the Motor domain to stabilize a particular composition, unlike myosin X for which interactions seems to make a particular position more favourable.

4. Fig. 3f: An arrow pointing to the barbed end is misleading. A double-headed arrow would be better or it should be mentioned that the arrow is illustrating the direction of the myosin movement.

The description of the arrow was added in the figure legend of this figure.

5. Fig. 3g: Which rigor-like MyoV pdb was used? Ref 42?

As stated in reference 42, the EM fitted rigor (42) and X-ray rigor-like (41) coordinates are very similar for the motor domain and the converter. Here we did use the rigor conformation.

6. Fig. 3h: As mentioned above, the accuracy of the angle should be treated with caution, as the pre-powerstroke state was shown to be quite flexible.

We agree that the angles of the powerstroke are just a measurement from structures and we have discussed in the paper how the powerstroke measured from structures do not reveal exact movement of the motor but provide an indication of the possible movement with additional flexibility to take into account, such as the pliant region and the exact position of the converter.

7. Fig. 3i: There is no red.

we have changed red for magenta in the legend

8. Fig. 3j: Low-pass filtering to indicate an overall position is used to emphasize a different position of the SH3 domain, but by performing this, it appears that the SH3 domain of the model does not fit the density. Furthermore, I would expect some density for the rest of the lever arm. Is it visible at another threshold without masking?

The SH3 subdomain in the rigor map is clearly much more mobile than other components of myosin V; therefore, it is not fully enclosed by the map boundary at the displayed contour level. We changed the wording in the Results section to better emphasize the apparent mobility of the SH3 domain, in order to address this point.

Our map actually shows density for most of the lever arm that is present in our truncated motor domain construct; the model in the paper includes up to residue 738 in the lever arm helix; the MD construct only goes to residue 741, so we are not missing too much from our density map.

9. Fig. 4: Which MyoV and how were the myosin motor domains superimposed?

The rigor coordinates are those from Myosin Va and the PPS from Myosin Vc. The motor domain were superimposed using the U50 and L50 subdomains involved in binding to F-actin. This was added to the figure legend.

This sentence is still there: "...reconstruction was $\sim 10 \text{ \AA}$ sufficient to unambiguously define the position and orientation of most alpha-helical elements in the map." I do not question that the model is correct or the fits are right. However, a map at around 10 \AA is definitely not high enough to unambiguously define the position and orientation of alpha-helices. I suggest to write "we could fit" rather than "unambiguously define the position and orientation".

We have double-checked the text to ensure the description of the resolution (9 \AA) is accurate everywhere. We also backed off on the description of the density quality- we now just say that most alpha helices were resolved.

The new method section describes the applied experiments and computational analyses in greater detail, which improved the quality of the manuscript significantly. Some points (especially on the model building of the rigor state of myosin X) need still to be clarified:

How was the myosin X MD in the rigor state created (page 22)? Homology modeling? It is only stated that 1OE9 was used. Afterwards, only "Fit in Map" (rigid-body) was used for the motor domain? If it is possible to "unambiguously define the position and orientation", then I would expect that a flexible fitting approach was used?

We added a phrase to the Methods text to indicate that we used Swiss Modeler to generate a homology model of the rigor motor domain. Flexible fitting was not used in this study.

Same paragraph: Which atomic model of F-actin was used?

We added text to indicate that PDB 3j8a was used for F-actin.

The fitting is procedure is still not clear, too. The authors state in the main text (page 10) that the converter domain was fitted as a rigid-body but in the method section they describe a more exhaustive way with the Situs package. Furthermore, they should add, when they only used a rigid-body fit (I guess, when they applied 'Fit in Map' in Chimera?).

Methods text has been amended to indicate that the Situs method is also a rigid-body search, but is more rigorous than the one in UCSF Chimera.

I do not understand, why two randomly chosen halves should result into two different conformations of the converter/lever arm position.

We added a sentence in the methods to clarify this point:

The resulting estimates of the converter position/orientation differ from one other; the extent of the differences reflects residual noise present in the reconstructions (noting that the residual noise in each half-data set refinement is essentially independent of the other half-data set refinement).

Please check the references in the Figure legends. I am a bit confused, since the authors state in one of the legends: "The parameters for the actin bundle were derived from ref 50". Is that correct?

This comes from a previous version. We have now corrected this. (ref 45).

It is obvious from Fig. 2b and d that the steps do not have uniquely defined sizes and more a likelihood distribution. This should be addressed as mentioned before because the bundles are not perfectly flat and ordered.

We do not feel that the data provides any insight as to the flatness or order of the bundles. A broad distribution of steps is seen even on a single filament, and is primarily a function of the motor, lever arm, and anti-parallel coiled coil allowing access to a number of actin binding sites. While the distribution is even broader on bundles, this is due to even more actin sites being available. The distribution does not imply anything about how well the bundles are ordered.